# Long Live The Balance:
# Information Bottleneck Driven Tree-based Policy Optimization

**Hao Jiang** [1]  **Shurui Li** [1]  **Tianpeng Bu** [1]  **Bowen Xu** [1]  **Xin Liu** [1]  **Qihua Chen** [1]  **Hongtao Duan** [1]  **Lulu Hu** [1]
**Bin Yang** [1]  **Minying Zhang** [1]

## Abstract

Recent advances in online reinforcement learning (RL) for large language models (LLMs) have demonstrated promising performance in complex reasoning tasks. However, they often exhibit an imbalanced exploration–exploitation trade-off, resulting in unstable optimization and sub-optimal performance. We introduce IB-Score, a novel metric grounded in Information Bottleneck theory that evaluates policy's exploration-exploitation balance by quantifying the trade-off between step-level reasoning diversity and mutual information shared with the correct answer. Analysis based on IB-Score shows that popular online RL approaches (*e.g.*, GRPO) with common regularizers fail to consistently maintain balance during training with suboptimal results. To address this, we propose **I**nformation **B**ottleneck-driven **T**ree-based **P**olicy **O**ptimization (**IB-TPO**), a principled framework that formulates IB-Score as a fine-grained optimization objective and utilizes a novel IB-guided tree sampling strategy that not only improves the efficiency of online sampling with 50% more trajectories under the same token budget, but also reuses the tree structure for effective IB-Score Monte Carlo estimation. Extensive experiments across standard benchmarks show that our method significantly outperforms GRPO baseline by 2.9% to 3.6% and also outperforms other state-of-the-art online RL approaches. Our code is available at Github.

## 1. Introduction

Recent state-of-the-art approaches in online reinforcement learning (RL) have propelled large language models (LLMs)

[1]Alibaba Cloud Computing, Alibaba Group. Correspondence to: Minying Zhang <minying.zmy@alibaba-inc.com>.

*Proceedings of the 43rd International Conference on Machine Learning*, Seoul, South Korea. PMLR 306, 2026. Copyright 2026 by the author(s).

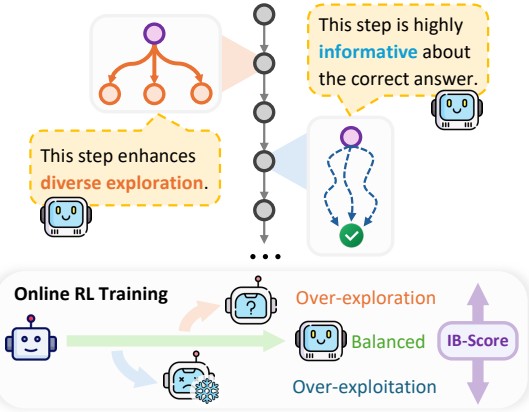

*Figure 1.* Striking an effective balance between exploration and exploitation remains a key challenge in online RL. We propose IB-Score to evaluate and encourage balanced policy optimization.

to achieve breakthrough performance in complex reasoning tasks such as mathematical reasoning (Luo et al., 2023; Shao et al., 2024; Wang et al., 2024), instruction following (Ren et al., 2025), function calling (Qian et al., 2025) and multimodal reasoning (Wei et al., 2025). Among them, Group Relative Policy Optimization (GRPO) (Shao et al., 2024) has emerged a leading approach but leaves fundamental challenges: (1) Inadequate exploration, as independent sampling strategy struggles to produce diverse trajectories with sufficient exploration due to structural inefficiency (Hou et al., 2025) and rapid model convergence (Cui et al., 2025; Cheng et al., 2025). (2) Insufficient exploitation, as sparse rewards from outcome-based verifiers limit fine-grained policy optimization (Li et al., 2025).

Existing approaches attempt to overcome these issues from different perspectives, *e.g.*, introducing entropy-based mechanisms to boost exploration (Cui et al., 2025; Cheng et al., 2025), employing process reward models to identify erroneous or critical reasoning steps (Shao et al., 2025), or incorporating tree structures to enable fine-grained reward estimation while strengthening exploration (Hou et al., 2025; Li et al., 2025). However, they still suffer from either the attenuation of learning signals due to policy's early convergence to a high-certainty local optimum (Yu et al., 2025; Xi et al., 2025) or training instability with degraded performance caused by exploding uncertainty (Zheng et al., 2025)

(Figure 1). Consequently, there is an urgent need for **a unified mechanism to quantify and encourage a balanced trade-off between exploration and exploitation** (Deng et al., 2026; Lei et al., 2025).

To address this, we diagnose the imbalanced online RL of LLMs from the perspective of Information Bottleneck (IB) theory (Tishby et al., 2000) in Section 3. We propose IB-Score, a novel metric that effectively quantifies the step-wise balance between policy exploration and exploitation. Specifically, IB-Score measures both diversity of policy's fine-grained exploration and the mutual information between each reasoning step and the correct answer. To investigate the relationship between IB-Score and model convergence, we conducted experiments to analyze the training dynamics of online RL baselines. Using IB-Score, we explain the underlying causes behind model's over-exploration and over-exploitation. Evidence suggests that maintaining a relatively stable IB-Score is beneficial for achieving sustained exploration diversity and higher performance.

Based on these, we propose **I**nformation **B**ottleneck driven **T**ree-based **P**olicy **O**ptimization (**IB-TPO**) in Section 4, which incorporates IB-Score into the RL objective to encourage the policy to adaptively strike a balance between exploration and exploitation throughout reasoning process. To this end, IB-TPO employs an IB-guided tree-based sampling strategy (IBTree) which not only selectively explores the model's most diverse and promising reasoning directions under the guidance of IB-Score, but also serves as an efficient Monte Carlo estimator of IB-Score. This mutual reinforcement between IB-Score estimation and tree-based sampling boosts exploration effectiveness of RL training.

We train our models at various scales of Qwen3 (Yang et al., 2025a) models and conduct comprehensive evaluations on multiple in/out-of-domain benchmarks. Experiments demonstrate that IB-TPO exhibits clear advantages by 2.9-3.6% over the GRPO baselines as well as state-of-the-art online RL methods. Moreover, our method not only achieves higher sampling effectiveness, but also maintains a better exploration–exploitation trade-off. This highlights the superiority of integrating IB with tree-based online RL.

In summary, our main contributions are threefold:

- We propose a novel metric IB-Score to quantify and diagnose the imbalanced exploration-exploitation trade-off in fine-grained level during LLM online RL.
- We propose IB-TPO, a novel tree-based policy optimization framework that incorporates IB-Score into the RL objective. IB-TPO leverages IBTree that enables IB-guided exploration and efficient IB-Score estimation.
- We conduct comprehensive experiments to demonstrate the superiority of our method over GRPO baselines and state-of-the-art approaches.

**Conflict of Interest Disclosure.** All authors of this paper are employed by Alibaba Cloud Computing, which leads the development of the Qwen3 series of models. The Qwen3-1.7B-Base and Qwen3-8B-Base models, as well as the Qwen3-14B-Base model used in the extended experiments, were among the models evaluated in this paper.

## 2. Preliminaries

**Group Relative Policy Optimization (GRPO)** (Shao et al., 2024) proposes a variant of Proximal Policy Optimization (PPO) (Schulman et al., 2017). Given problem $q$ and the policy model $\pi_\theta$ parameterized by $\theta$, GRPO independently samples $G$ trajectories $\{\tau_i\}_{i=1}^{G}$ from $\pi_\theta(\cdot|q)$ to explore different ways of solving problem. Then, reward signals $\{R_i\}_{i=1}^{G}$ are collected from verifier (*e.g.*, outcome math verifier, code executor) and normalized in group to calculate relative advantages compared to group baseline. Finally, the policy optimization with clipping strategy is performed as follows:

$$
J_{\text{GRPO}}(\theta) = \mathbb{E}_{\tau \sim \pi_\theta(\cdot|q)} \frac{1}{G} \sum_{i=1}^{G} \frac{1}{|\tau_i|} \sum_{t=1}^{|\tau_i|}
$$

$$
\left[ \min\left(w_{i,t}(\theta)A_{i,t}, \text{clip}(w_{i,t}(\theta), 1-\epsilon, 1+\epsilon)A_{i,t}\right) \right. \quad (1)
$$

$$
\left. - \beta_{KL} D_{KL}(\pi_\theta || \pi_{\text{ref}}) \right],
$$

where

$$
w_{i,t}(\theta) = \frac{\pi_\theta(\tau_{i,t}|q, \tau_{i,<t})}{\pi_{\text{ref}}(\tau_{i,t}|q, \tau_{i,<t})}, \ A_{i,t} = \frac{R_i - \text{mean}(\{R_i\}_{i=1}^{G})}{\text{std}(\{R_i\}_{i=1}^{G})},
$$

$$(2)$$

denote the importance weight and advantage of each trajectory token $\tau_{i,t}$, respectively.

**Information Bottleneck (IB) Theory** (Tishby & Zaslavsky, 2015; Tishby et al., 2000) provides a theoretical framework for extracting relevant information from an input variable $X$ to predict an output variable $Y$ through an intermediate representation $Z$. The objective is formulated as:

$$
\min I(X; Z) - \beta I(Z; Y), \quad (3)
$$

where $I(\cdot; \cdot)$ denotes mutual information. Specifically, minimizing $I(X; Z)$ encourages model generalization by discarding task-irrelevant information, while maximizing $I(Z; Y)$ promotes the relevance between the intermediate representation and the target output. This dual objective offers a unified perspective for balancing the trade-off between exploration and exploitation of the model. In practice, to make this objective tractable, the mutual information terms are often expanded using entropy $H(\cdot)$: $I(X; Y) = H(X) - H(X|Y)$, serving as the mathematical foundation for IB objective.

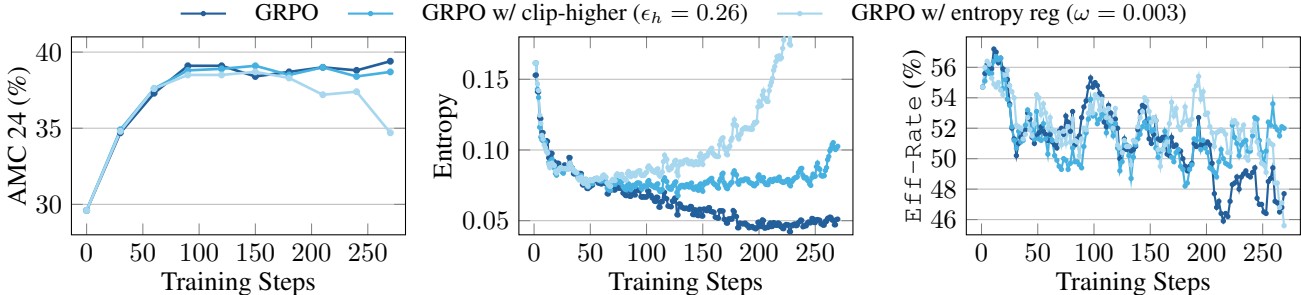

*Figure 2.* Training dynamics of GRPO baselines using Qwen3-8B-Base. `Eff-Rate` denotes the proportion of effective sampled groups with non-zero reward variance. Both clip-higher and entropy regularization fail to improve performance.

## 3. Diagnosing Exploration-Exploitation Imbalance: An IB Perspective

### 3.1. Exploration-Exploitation Dilemma in Online RL

To uncover the challenges in exploration-exploitation balancing of online RL, we employ Qwen3-8B-Base and conduct experiments with GRPO baseline and its variants with two commonly used regularization methods: clip-higher and entropy regularization.

As shown by the training dynamics in Figure 2, the GRPO baseline suffers from performance stagnation. This issue stems from premature convergence in the early stages of training, where a rapid decrease in policy entropy causes the model to become overly deterministic (Cui et al., 2025). This leads to to a sharp decline in the proportion of effective sampled groups with non-zero reward variance (termed `Eff-Rate`) and results in sparse learning signal for policy optimization and plateaued validation accuracy. Although techniques like asymmetric clipping thresholds and entropy regularization can alleviate entropy degradation, they fail to improve model performance. Despite the increased entropy, their `Eff-Rate` continues to decline throughout training, indicating limited exploration effectiveness. Moreover, naively encouraging higher entropy often leads to entropy explosion (where the model exhibits excessive uncertainty) and results in training instability.

This highlights a fundamental dilemma in current online RL approaches: they not only struggle to incentivize effective exploration, but also lack robust mechanism to balance exploration and exploitation. To address this challenge, we turn to the Information Bottleneck (IB) theory by proposing it as a new lens to quantify and navigate this trade-off.

### 3.2. Quantifying Exploration–Exploitation via IB

Although existing approaches (Deng et al., 2026; Lei et al., 2025) have demonstrated the superiority of the applying IB theory to LLM reasoning and reinforcement training, they still lack an effective and efficient way for leveraging IB to quantify and encourage the fine-grained exploration-

exploitation balance in online RL. In this section, we aim to reformulate IB as an interpretable quantitative metric to evaluate this trade-off in a fine-grained manner.

Given a policy model $\pi_\theta$, a problem $q$, a multi-step reasoning process $\tau = \{s_i\}_{i=1}^{N_\tau} \sim \pi_\theta(\cdot|q)$ and the ground-truth answer $a^*$, we reformulate the objective of the information bottleneck from Equation 3 as maximizing:

$$
\begin{aligned}
J_{\text{IB}}(\tau) &= -I(q;\tau) + \beta I(\tau; a^*|q) \\
&= -[H(\tau) - H(\tau|q)] + \beta[H(\tau|q) - H(\tau|a^*, q)] \quad (4) \\
&= -H(\tau) + (\beta+1)H(\tau|q) - \beta H(\tau|a^*, q).
\end{aligned}
$$

Since $H(\tau)$ is a problem-agnostic marginal entropy while RL post-training aims to enhance policy's problem-solving capability (*i.e.*, optimizing $\pi_\theta(\cdot|q)$), we follow prior work (Lei et al., 2025; Fischer, 2020) to remove this term and simplify $J_{\text{IB}}(\tau)$ into a task-specific objective:

$$
\begin{aligned}
J_{\text{IB}}(\tau) &= (\beta+1)H(\tau|q) - \beta H(\tau|a^*, q) \\
&= \sum_{i=1}^{N_\tau} \left[ (\beta+1)\underbrace{H(s_i|q, s_{<i})}_{\text{exploration term}} - \beta \underbrace{H(s_i|a^*, q, s_{<i})}_{\text{exploitation term}} \right], \quad (5)
\end{aligned}
$$

where we transform the entropy terms into step-wise formulation, given the autoregressive nature of LLM prediction.

**Derivation 1** (Estimation of $H(s_i|q, s_{<i})$). Previous approaches have attempted to encourage token-level entropy to mitigate sharpening of the distribution (Cui et al., 2025; Cheng et al., 2025). However, this strategy often amplifies the role of token selection in model decisions and risks causing entropy explosion (Huang et al., 2025). Although advanced works have pointed out that model often makes decisions at the step level (Bu et al., 2025; Yao et al., 2023), obtaining the entire probability distributions of reasoning steps is impractical. To overcome this, we treat $B$ candidates $s_i^b$ generated from shared prefix $(q, s_{<i})$ as Monte Carlo (Metropolis & Ulam, 1949) samples from $\pi_\theta(\cdot|q, s_{<i})$ and calculate the Tsallis entropy (Tsallis, 1988) as a tractable surrogate of Shannon entropy (Shannon, 1948)

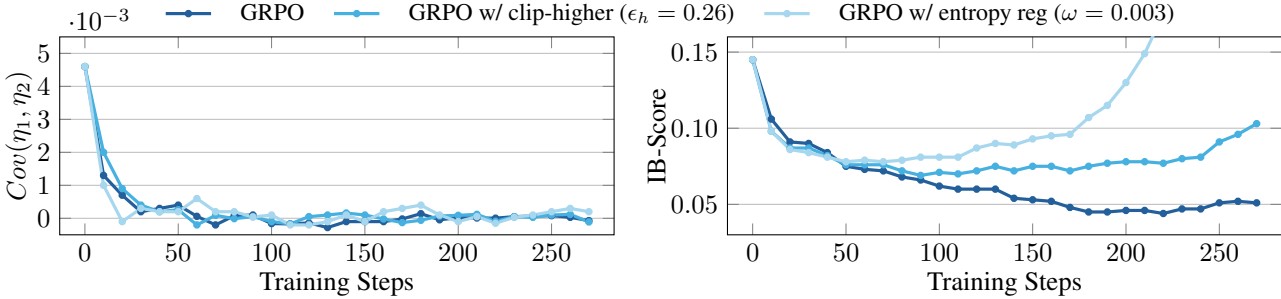

*Figure 3.* Training dynamics of $Cov(\eta_1, \eta_2)$ and IB-Score for GRPO baselines. We use Qwen3-8B-base and $\beta = 5$.

to improve numerical stability under sparse sampling:

$$H(s_i|q, s_{<i}) := \mathbb{E}_{s_i \in \pi(\cdot|q, s_{<i})} \left[ \frac{1 - \pi_\theta(s_i|q, s_{<i})^{\alpha-1}}{\alpha - 1} \right]$$

$$\approx \frac{1}{\alpha - 1} \left[ 1 - \frac{1}{B} \sum_{b=1}^{B} \pi_\theta(s_i^b|q, s_{<i})^{\alpha-1} \right]$$

(6)

as $\hat{H}(s_i|q, s_{<i})$, where $\alpha \in \mathbb{R}$ is the entropic index that we empirically set as 2. To align steps with different lengths and provide robust to outliers (Zhao et al., 2025b), we normalize the step probability by geometric average:

$\pi_\theta(s_i|q, s_{<i}) = \left[ \prod_{t=1}^{|s_i|} \pi_\theta(s_{i,t}|q, s_{<i}, s_{i,<t}) \right]^{\frac{1}{|s_i|}}$, where $\pi_\theta(s_{i,t}|q, s_{<i}, s_{i,<t})$ is the token-wise probability extracted from outputting logits.

**Derivation 2** (Estimation of $H(s_i|a^*, q, s_{<i})$). Intuitively, $H(s_i|a^*, q, s_{<i})$ reflects the uncertainty of step $s_i$ occurring given the prediction of correct answer, and measures the informative contribution that $s_i$ offers to problem solving (Lei et al., 2025). Similar to Equation 6, we estimate it via:

$$\hat{H}(s_i|a^*, q, s_{<i}) = 1 - \frac{1}{B} \sum_{b=1}^{B} p(s_i^b|a^*, q, s_{<i}), \quad (7)$$

where the posterior probability $p(s_i|a^*, q, s_{<i})$ is conditioned on prefix $(q, s_{<i})$ and ground-truth answer $a^*$, it represents the probability of predicting the current step $s_i$ given the premise that the correct answer is accessed. We can convert it using Bayes Formula:

$$p(s_i^b|a^*, q, s_{<i}) = \frac{p(a^*|q, s_{<i}, s_i^b) \cdot \pi_\theta(s_i^b|q, s_{<i})}{p(a^*|q, s_{<i})}, \quad (8)$$

where the reward density $p(a^*|q, s_{<i})$ denotes the probability of reaching correct answer (*i.e.*, obtain reward) given prefix $(q, s_{<i})$. In practice, we perform Monte Carlo estimation on $p(a^*|q, s_{<i})$ by multiple rollouts starting from the prefix $(q, s_{<i})$: $\hat{p}(a^*|q, s_{<i}) := \frac{1}{|\mathcal{T}^{(s_{<i})}|} \sum_{\tau \in \mathcal{T}^{(s_{<i})}} R(\tau)$, where trajectories $\mathcal{T}^{(s_{<i})} = \{\tau | \tau \sim \pi_\theta(\cdot|q), s_{<i} \prec \tau\}$ and $R(\cdot) \in [0, 1]$ denotes the reward function.

For notational convenience, we will omit the prefix $(q, s_{<i})$ in subsequent mathematical expressions and define the following simplified expressions:

> **Symbol Simplification**
>
> $\pi_\theta(s_i)$ : step probability $\pi_\theta(s_i|q, s_{<i})$
> $p(s_i|a^*)$ : step posterior probability $p(s_i|a^*, q, s_{<i})$
> $p(a^*|s_i)$ : step reward density $p(a^*|q, s_{<i+1})$

The same for entropy $H(s_i|a^*)$ and $H(a^*|s_i)$.

**Information Bottleneck Score (IB-Score).** We summarize the Derivations 1 and 2, then reformulate the IB objective for step $s_i$ as:

$$J_{\text{IB}}(s_i) \approx (\beta + 1)\hat{H}(s_i) - \beta\hat{H}(s_i|a^*)$$

$$= 1 + \frac{\beta}{B} \sum_{b=1}^{B} \eta_1(s_i^b) \cdot \eta_2(s_i^b), \quad (9)$$

and

$$\eta_1(s_i^b) = -(1 + \frac{1}{\beta}) + \frac{\hat{p}(a^*|s_i^b)}{\hat{p}(a^*|s_{i-1})},$$

$$\eta_2(s_i^b) = \pi_\theta(s_i^b), \quad (10)$$

where $\eta_1$ quantifies the information gain from environmental feedback and $\eta_2$ represents model's confidence in each branch. The interplay of these two terms allows $J_{\text{IB}}(s_i)$ to function as a comprehensive metric, and we term it as the **I**nformation **B**ottleneck **Score** (**IB-Score**). Further analysis in Appendix A reveals that the IB-Score offers a more nuanced objective than simply measuring the model's uncertainty (*i.e.*, entropy), as it critically dependent on the covariance $Cov(\eta_1, \eta_2)$. This implies that achieving an effective exploration-exploitation balance requires more than just **encouraging high overall uncertainty**. Instead, the model must learn to **strategically allocate its confidence to branches where the feedback is most informative**, thus creating a positive correlation between $\eta_1$ and $\eta_2$.

### 3.3. Diagnosing Imbalanced Online RL from IB

To elucidate imbalanced online RL, we track training dynamics through the lens of the IB in Figure 3 as supplement to experiments in Section 3.1 (see Section 5.1 for detailed setup). As illustrated, the model exhibits a natural initial positive covariance, indicating its intrinsic ability to align confidence with reasoning paths that offer high information gain

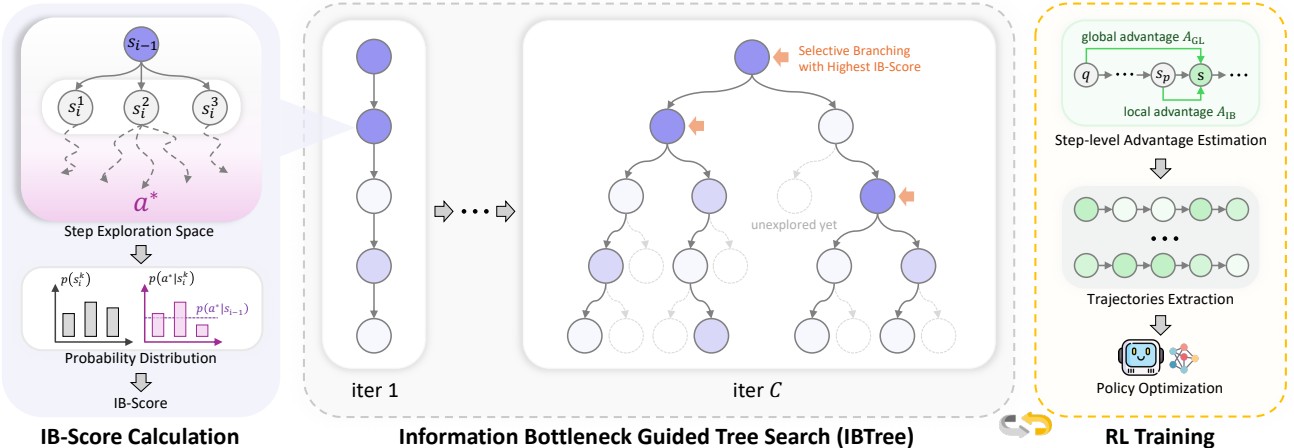

*Figure 4.* Overview of our method. We initialize IBTree with a given problem, then perform tree expansion in multiple iterations. In each iteration, we calculate the IB-Score of each node based on existing rollouts from it, and selectively branching at nodes with highest IB-Scores to improve exploration efficiency and effectiveness. After expansion, we compute global advantage and IB-based local advantage for each node, then optimize the policy model using step-level advantages.

to correct answer. However, as training progresses, we find that GRPO baselines suffer a rapid decline in $Cov(\eta_1, \eta_2)$ until it approaches zero, implying that the model's confidence becomes uniform and less distinguishable. This dynamic raises a conflict between goals of maintaining high confidence in high-value reasoning paths and preserving sufficient exploration capability, making regularization terms yield either marginal improvement or training collapse.

# 4. Information Bottleneck Driven Tree-based Policy Optimization

In Section 3, we propose the IB-Score to evaluate online RL training and demonstrate the existing exploration-exploitation balancing issue in current online RL approaches. To address this issue, we incorporate the IB-Score into the optimization objective of online RL. However, this is non-trivial, as estimating IB-Score on every step will significantly increase the burden of online sampling and reduce training efficiency. To overcome this, we introduce an efficient tree sampling structure in Section 4.1 that enhances sampling efficiency and enables effective estimation of the IB-Score. And then perform IB-guided online RL with structured sampled trajectories in Section 4.2.

## 4.1. Information Bottleneck Guided Tree Search

Although previous approaches (Li et al., 2025; Yang et al., 2025b) introduce tree structured sampling into online RL of LLMs, they suffer from enormous search space and degraded efficiency. TreeRL (Hou et al., 2025) introduces an entropy-guided tree search to improve exploration efficiency by selectively branching on tokens where the model exhibits the highest uncertainty. However, this strategy ignores environmental feedback and there is a misalignment between

**Algorithm 1** IBTPO

---

**Input**: Problem $q$, Policy $\pi_\theta$, Ref Policy $\pi_{\text{ref}}$, Branching Iterations $L$, Branching Nodes $K$, Branches $B$
**Output**: Optimized Policy $\pi_\theta^*$
    // IBTree Initialization
1: $\mathcal{T} \leftarrow \{\}, \mathcal{S} \leftarrow \{q\}$
    // IBTree Expansion
2: **for** $l = 1$ to $L$ **do**
3:     $\mathcal{S}' \leftarrow \arg \text{topK}_{s \in \mathcal{S}} (\text{IB-Score}(s))$
4:     $\mathcal{T}' \leftarrow \bigcup_{k=1}^{|\mathcal{S}'|} \bigcup_{b=1}^{B} \{\tau_b | \tau_b \sim \pi_\theta(\cdot|q), \text{pre}(\mathcal{S}'_k) \prec \tau_b\}$
5:     $\mathcal{T} \leftarrow \mathcal{T} \cup \mathcal{T}'; \mathcal{S} \leftarrow \mathcal{S} \cup \text{nodes}(\mathcal{T}')$
6: **end for**
    // Step-level Advantage Calculation
7: **for** each step $s$ in the tree nodes $\mathcal{S}$ **do**
8:     Calculate local advantage $A_{\text{IB}}(s)$ by Equation 14
9:     Calculate global advantage $A_{\text{GL}}(s)$ by Equation 15
10:     $A(s) \leftarrow A_{\text{IB}}(s) + \lambda \cdot A_{\text{GL}}(s)$
11: **end for**
    // Policy Optimization
12: $\pi_\theta^* \leftarrow$ Update $\pi_\theta$ by Equation 1 with advantage $A(s)$

---

token-level entropy and the model's decision-making.

Based on these, we propose Information Bottleneck Guided Tree Search (**IBTree**) as illustrated in Figure 4, which selectively branches with the highest IB-Score, as higher IB-Score indicates higher exploration diversity with promising confidence allocation. This IB-guided branching strategy enables tree search to achieve better exploration efficiency. We decompose the tree sampling process into $L$ iterations, where in each iteration we expand the tree from $K$ nodes with highest IB-Scores, each of them generates $B$ new trajectories. We fix $K = 1$ for the first iteration (only the root

node is expandable) with $B_0$ initial trajectories, so the total number of trajectories is $G = B_0 + (L-1) \times K \times B$.

**Initialization.** For each problem $q$, we initialize a tree with only the root node $q$. The set of trajectories $\mathcal{T}$ and the set of of tree nodes $\mathcal{S}$ can be expressed as:

$$\mathcal{T} = \{\}, \ \mathcal{S} = \{q\}. \tag{11}$$

**Expansion.** In each iteration, we sort tree nodes by IB-Score in descending order and select top-K for branching:

$$\mathcal{S}' = \arg \underset{s \in \mathcal{S}}{\mathrm{topK}} \, (\text{IB-Score}(s)), \tag{12}$$

where $\mathcal{S}' = \mathcal{S}$ if $|\mathcal{S}| \le K$, and we avoid branching after leaf nodes to ensure the validity. Then, we continue reasoning generation from each branching node by $B$ times:

$$\mathcal{T}' = \bigcup_{k=1}^{|\mathcal{S}'|} \bigcup_{b=1}^{B} \{\tau_b | \tau_b \sim \pi_\theta(\cdot|q) \text{ s.t. } \text{pre}(\mathcal{S}'_k) \prec \tau_b\}, \tag{13}$$

where $\text{pre}(\mathcal{S}'_k)$ denotes the shared prefix response up to node $\mathcal{S}'_k$ that has already been generated in previous iterations. We use \n\n to delimit steps (Xiao et al., 2025) in each trajectory. Then we update $\mathcal{T}$ by $\mathcal{T} = \mathcal{T} \cup \mathcal{T}'$ and $\mathcal{S}$ by $\mathcal{S} = \mathcal{S} \cup \left( \bigcup_{j=1}^{|\mathcal{T}'|} \{s|s \in \mathcal{T}'_j\} \right)$.

Compared to independent sampling, IBTree achieves higher sampling efficiency by sharing prefix responses during branching and enables more attempts under same token budget. Compared to (Li et al., 2025; Yang et al., 2025b), IBTree leverages IB-Score to selectively branch and explore on most promising directions. Compared to TreeRL (Hou et al., 2025), IBTree provides a more robust and effective guidance for branching node selection. Furthermore, IB-Tree serves as Monte Carlo estimator of IB-Score and can adapt branching node selection as IB-Score updating. We compare different sampling strategies in Table 3.

### 4.2. Online RL Training with IBTree

In each training iteration, we sample $G$ trajectories with fine-grained IB-Scores for each problem using IBTree, then collect supervision signals base on the tree-structured search.

To encourage balanced exploration-exploitation trade-off, we incorporate IB-Score of reasoning steps into optimization objective. Specifically, we transform Equation 9 into a surrogate objective that maximizes $\eta_1(s) \cdot \eta_2(s)$ for each sampled step $s \in \mathcal{S}$ ($s \ne q$), which is formulated as:

$$\tilde{J}_{\text{IB}}(s) = \eta_1(s) \cdot \eta_2(s) = A_{\text{IB}}(s) \cdot w(s),$$
$$A_{\text{IB}}(s) = \left[ \frac{\hat{p}(a^*|s)}{\hat{p}(a^*|s_p)} - \left(1 + \frac{1}{\beta}\right) \right] \cdot \pi_{\text{ref}}(s), \tag{14}$$

where $s_p$ is the parent of $s$, $w(s) = \pi_\theta(s)/\pi_{\text{ref}}(s)$ denotes importance weight and we transform IB objective to a standard policy gradient term in step-level. Notably, $A_{\text{IB}}(s)$ serves as the local advantage of $s$ relative to $s_p$. As a complement, we introduce a global advantage as follows:

$$A_{\text{GL}}(s) = \frac{\hat{p}(a^*|s) - \hat{p}(a^*|q)}{\text{std}(\{R(\tau)\}_{\tau \in \mathcal{T}})}, \tag{15}$$

where $\hat{p}(a^*|q)$ presents reward density of the root. Then, the final advantage is combination of local and global terms:

$$A(s) = A_{\text{GL}}(s) + \lambda \cdot A_{\text{IB}}(s), \tag{16}$$

where $\lambda$ denotes the weight of advantage $A_{\text{IB}}(s)$. Finally, We employ policy gradient objective in Equation 1 for RL. See Algorithm 1 for overall pipeline of our method.

## 5. Experiments

### 5.1. Setup

**Models and Dataset**. Our experiments are mainly conducted on Qwen3-1.7B-Base and Qwen3-8B-Base (Yang et al., 2025a), a popular model family that widely employed in RL researches and cover different parameter scales. We select DAPO-Math-17K dataset (Yu et al., 2025) as our training dataset, which contains about 17K challenging mathematical problems paired with ground-truth answers.

**Training Details.** We implemented our training framework on ms-swift (Zhao et al., 2025a) which employs vLLM (Kwon et al., 2023) with prefix caching technology for inference acceleration. During the sampling stage, we sample trajectories with recommended parameters T = 0.7, top_p=0.95, top_k=20 and a maximum truncation length of 2K per trajectory. For each training iteration, we sample 128 problems for generating trajectories. For independent sampling approaches, we sample 8 trajectories for each problem, and for tree-based approaches we sample 12 trajectories for each problem with $(B_0, L, K, B) = (4, 9, 1, 1)$ which have equivalent token consumption to that of 8 independent trajectories (see Table 3). We set the learning rate to $1 \times 10^{-6}$, the KL regularization weight to 0.001, and we train the model for 1 epoch. All of our experiments are conducted on 8×A100 GPUs. Please see Appendix B.1 for more implementation details for our method.

**Benchmark Evaluation.** We employ EvalScope (Team, 2024) to evaluate the performance of our LLMs. We evaluate models on several widely used challenging mathematical benchmarks: MATH-500 (Lightman et al., 2023), AIME 24/25, AMC 23/24 and two out-of-domain benchmarks: GPQA Diamond (Rein et al., 2024) and IFEval (Zhou et al., 2023). To ensure the reliability of the results, we employ **avg@32** as low-variance metric in our comparisons.

*Table 1.* Main comparison with state-of-the-art methods. We report `avg@32` accuracy for every benchmark, the best results are bolded. [†] Since entropy regularization with $\omega = 0.003$ fails to achieve comparable results, we carefully use $\omega = 0.001$ for comparison.

| Models | Mathematics | | | | | Multi-Task | Instruction | |
|---|---|---|---|---|---|---|---|---|
| | MATH-500 | AIME 24 | AIME 25 | AMC 23 | AMC 24 | GPQA | IFEval | Overall |
| **Qwen3-1.7B-Base** | | | | | | | | |
| Initial Model | 51.2% | 3.8% | 1.7% | 24.5% | 15.1% | 21.2% | 21.9% | 19.9% |
| - Vanilla GRPO | 66.8% | 7.0% | 4.5% | 35.6% | 19.7% | 26.5% | 24.0% | 26.3% |
| - GRPO w/ Clip-higher | 65.8% | 7.7% | 4.5% | 35.0% | 20.5% | 24.8% | 23.7% | 26.0% |
| - GRPO w/ Entropy Reg[†] | 64.8% | 6.6% | 2.2% | 33.6% | 18.8% | 24.7% | 23.7% | 24.9% |
| - IBRO (Lei et al., 2025) | 66.4% | 5.6% | 4.1% | 34.6% | 20.8% | 24.7% | 25.4% | 25.9% |
| - TreeRL (Hou et al., 2025) | 67.2% | 7.9% | 4.6% | 37.0% | 20.6% | 26.8% | 23.5% | 26.8% |
| - TreePO (Li et al., 2025) | 66.6% | 7.2% | 4.5% | 36.5% | 19.5% | 26.2% | 23.2% | 26.2% |
| - IBTPO (Ours) | **70.1%** | **9.5%** | **6.7%** | **38.7%** | **23.4%** | **29.0%** | **26.9%** | **29.2%** |
| **Qwen3-8B-Base** | | | | | | | | |
| Initial Model | 65.7% | 11.5% | 8.9% | 41.7% | 26.8% | 26.1% | 43.8% | 32.0% |
| - Vanilla GRPO | 81.5% | 17.1% | 13.6% | 53.5% | 39.4% | 38.1% | 42.0% | 40.7% |
| - GRPO w/ Clip-higher | 81.2% | 16.8% | 13.5% | 53.3% | 38.9% | 37.8% | 45.4% | 41.0% |
| - GRPO w/ Entropy Reg[†] | 81.8% | 17.0% | 14.0% | 54.3% | 39.4% | 39.1% | 44.2% | 41.4% |
| - IBRO (Lei et al., 2025) | 82.0% | 16.9% | 14.5% | 55.3% | 39.5% | 38.5% | 44.7% | 41.6% |
| - TreeRL (Hou et al., 2025) | 82.5% | 17.8% | 14.9% | 56.1% | 40.5% | 39.8% | 42.5% | 42.0% |
| - TreePO (Li et al., 2025) | 82.2% | 17.5% | 14.7% | 55.8% | 40.1% | 39.2% | 42.2% | 41.6% |
| - IBTPO (Ours) | **83.3%** | **19.7%** | **15.3%** | **57.9%** | **46.0%** | **41.7%** | **46.2%** | **44.3%** |

**IB-Score Evaluation.** We employ a standard offline pipeline to evaluate and compare exploration–exploitation balance across different models through IB-Score. Specifically, we sample 4 seed trajectories for each problem in a random subset from DAPO-Math-17K with 1024 problems, perform 5 rollouts from every non-terminal steps to calculate IB-Scores, and finally average across all steps.

### 5.2. Main Results

**Baselines.** We compare our method with GRPO baseline as well as various representative state-of-the-art approaches. Specifically, in addition to the GRPO baseline and its variants incorporating clip-higher and entropy regularization, we also compare with IBRO (Lei et al., 2025), TreeRL (Hou et al., 2025) and TreePO (Li et al., 2025). IBRO transforms IB objective as a sequence-level advantage-weighted entropy regularization term, TreeRL and TreePO introduce efficient tree search into online RL. To ensure a fair comparison, we evaluate them using officially provided open-source codes with recommended parameters, and align the basic experimental settings. We don't include concurrent work IIB-LPO (Deng et al., 2026) in our comparison, as it proposes a regularization term similar to IBRO and has not yet released code. Please see Appendix B.2 for more implementation details of compared approaches.

**Main Comparison.** As shown in Table 1, our method significantly outperforms the GRPO baseline, achieving an average improvement of 2.9% and 3.6%, and also surpasses all compared methods across different benchmarks. We

*Table 2.* Ablation of main components of our method.

| Models | AIME 25 | AMC 24 | GPQA |
|---|---|---|---|
| TreeRL (w/ EPTree) | 14.9% | 40.5% | 39.8% |
| Vanilla GRPO | 13.6% | 39.4% | 38.1% |
| + IBTree | 15.0% | 43.8% | 40.8% |
| + IBTPO Adv (Eq 16) | 14.2% | 42.5% | 41.2% |
| + RandTree & IBTPO Adv | 14.5% | 39.8% | 37.3% |
| + EPTree & IBTPO Adv | 15.0% | 42.3% | 40.9% |
| + IBTree & IBTPO Adv (Ours) | **15.3%** | **46.0%** | **41.7%** |

also compare the training dynamics in Figure 5, where our method achieves the most consistent training improvement on the validation datasets. Additionally, our tree sampling method significantly outperforms independent sampling in terms of `Eff-Rate` (by average of 6.8%), demonstrating the superior exploration diversity of IBTree. Moreover, benefiting from the fine-grained IB objective, our method shows significantly better dynamics on $Cov(\eta_1, \eta_2)$ and IB-Score compared to other approaches. This indicates that our method not only achieves higher exploration diversity but also maintains effective confidence allocation capability. We also report **pass@K** accuracy in Appendix C.2.

### 5.3. More Analysis

**Effectiveness of main components.** Our method consists of two main components: IBTree and IB-based advantage estimation. We conduct ablation study in Table 2, where we use GRPO's independent sampling and the advantage

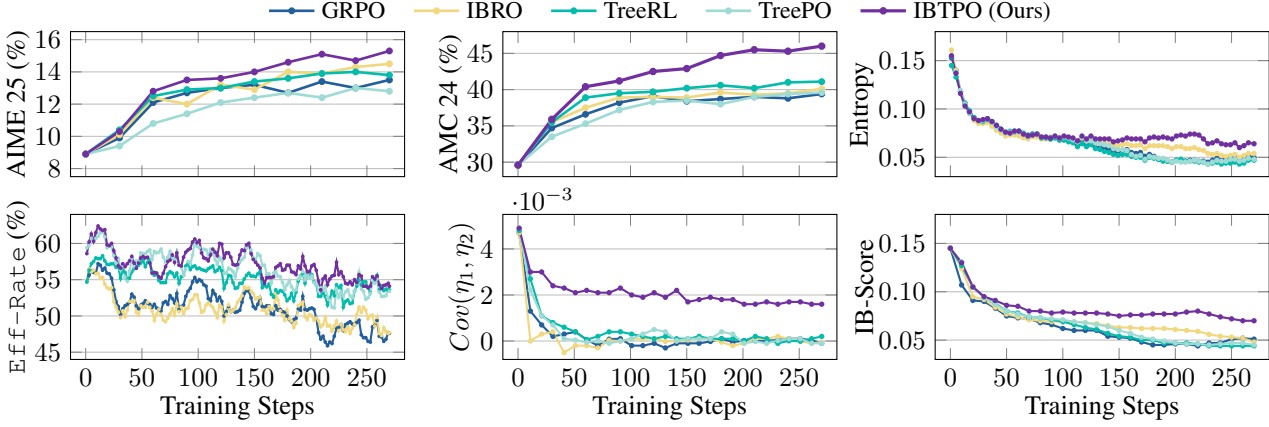

*Figure 5.* Training dynamics of our method compared with state-of-the-art approaches using Qwen3-8B-base.

*Table 3.* Ablation of different tree branching strategies.

| Branching Strategy | G | $\beta$ | Eff-Rate↑ | Avg-Rate↑ | #Tokens↓ |
|---|---|---|---|---|---|
| no branching | 8 | - | 54.7% | 19.6% | **7,469** |
|  | 12 | - | 59.8% | 20.1% | 12,035 |
| random | 12 | - | 48.4% | 20.0% | 7,579 |
| fix-width | 12 | - | 59.4% | 19.9% | 8,306 |
| entropy guided | 12 | - | 57.8% | 21.6% | 7,784 |
| IB-Score guided | 12 | 1.0 | 60.0% | 22.3% | 7,663 |
|  | 12 | 5.0 | **60.2%** | **23.2%** | 7,592 |
|  | 12 | 10.0 | 59.3% | 22.0% | 7,824 |

*Table 4.* Ablation of parameter $\lambda$ in advantage estimation.

| $\lambda$ | AIME 25 | AMC 24 | GPQA |
|---|---|---|---|
| 0 (only global advantage) | 15.0% | 43.8% | 40.8% |
| 0.05 | 14.9% | 45.6% | **41.9%** |
| 0.1 | **15.3%** | **46.0%** | 41.7% |
| 0.5 | 13.7% | 42.0% | 39.8% |

*Table 5.* Ablation of step delimiter.

| Models | AIME 25 | AMC 24 | GPQA |
|---|---|---|---|
| w/o random noise | 15.3% | 46.0% | 41.7% |
| w/ random noise | 15.5% | 45.7% | 41.5% |

estimation in Equation 2 as alternative baseline strategies. The results in the table reveal the following observations: (1) Both components improve GRPO's performance. (2) EPTree & IBTPO Adv outperforms TreeRL, demonstrating that the IB-based objective yields a non-negligible improvement. (3) The full combination (IBTree & IBTPO Adv) achieves the best results, confirming that IBTree guided by IB-Score cannot be readily replaced by alternative tree structures and further highlighting the synergistic nature of our framework.

**Effectiveness of IBTree.** To demonstrate the effectiveness of IBTree, we compare it with several different sampling strategies in Table 3: (1) Independent sampling with `no branching`, (2) `random` branching, (3) branching at every step until the tree width reaches the target `fix-width` G (Li et al., 2025), (4) `entropy guided` (Hou et al., 2025) branching with most uncertain token, and (5) `IB-Score guided` branching (our strategy). We use Qwen3-8B-Base on a random sampled subset of DAPO-Math-17K dataset with 1024 problems for analysis. More parameters are compared in Appendix C.1.

Although `random` and `fix-width` save token cost, they suffer from reduced `Eff-Rate` since shared reasoning prefixes naturally reduces the diversity between trajectories. The `entropy guided` branching strategy improves diversity and `Eff-Rate` by selectively branching at high-

entropy tokens. Our `IB-Score guided` leverages IB-Score as branching guidance, achieving the best `Eff-Rate` and improving the overall accuracy (`Avg-Rate`). Moreover, under similar token budget as independent sampling, our tree sampling strategy generates 50% more trajectories.

We also compare GPU wall-clock runtime in Appendix C.3. Notably, we introduce implementation optimizations to improve the efficiency of IBTree, enabling it to achieve better performance while attaining higher runtime efficiency than independent sampling.

**Ablation of parameter $\beta$.** The parameter $\beta$ controls the weights of two terms in Equation 5. We compare different $\beta$ in Table 3, the results show that $\beta = 5$ produces better `Eff-Rate` and `Avg-Rate` with less token consumption.

**Ablation of parameter $\lambda$.** We conduct an ablation study of advantage weight $\lambda$ in Equation 16. As shown in Table 4, training with only global advantage achieves sub-optimal results and $\lambda = 0.1$ achieves the best performance.

**Ablation of step delimiter.** We use `\n\n` as step delimiter in our framework, which is a simple, training-free, and natural formatting boundary in chain-of-thought responses. To validate its robustness, we additionally conduct a noise-

injection experiment in Table 5, where we perturb 10% of existing \n\n split points by relocating them to random positions to simulate under-segmentation and over-segmentation noise. As shown, the performance change is not significant, indicating that our method is robust to moderate step-boundary perturbations.

## 6. Related Work

### 6.1. Online RL for LLM Reasoning

Previous advances (Guo et al., 2025a; Shao et al., 2024; Schulman et al., 2017) have demonstrated the superiority of online reinforcement learning (RL) in large language models (LLMs) post-training. Although promising, recent approaches (Yue et al., 2025; Deng et al., 2025; Yu et al., 2025) indicate that balanced exploration-exploitation benefits RL training. To this end, recent work (Deng et al., 2026; Lei et al., 2025) propose advantage-weighted entropy regularization terms based on the IB to encourage this balance. Process supervision are also proved to be an effective way to enhance LLM reasoning (Lightman et al., 2023) and online RL (Zhang et al., 2025; Park et al., 2025), but it remains a key challenge to efficiently collect process supervision.

### 6.2. Tree Search in Online RL

Recent works (Qi et al., 2024; Zhang et al., 2024; Hao et al., 2023) have demonstrated that tree search enhances the reasoning capabilities of LLMs, as it not only extends the depth of reasoning through structured exploration but also leverages tree structures to provide process supervision. (Yang et al., 2025b; Ji et al., 2025; Guo et al., 2025b) further integrate tree search into online RL. However, most of them rely on large tree search space, hindering this integration. To address this, TreeRL (Hou et al., 2025) introduces an entropy-guided branching strategy to boost exploration of tree searching in limited token budget, and TreePO (Li et al., 2025) controls sampling cost with limited tree width.

## 7. Conclusion

This work addresses the imbalanced exploration-exploitation in online RL. We introduce fine-grained metric IB-Score to evaluate this trade-off and propose IB-TPO, a unified framework integrating IB-Score into policy optimization with IBTree. Comprehensive experiments demonstrate the superiority of IB-TPO, highlighting the promise of integrating IB with tree-based online RL.

## 8. Limitation

Despite effective, we still need to address the time consumption introduced by multi-iteration tree sampling. Although parallel sampling multiple trees can fully utilize decoding width of inference engine, IBTree remains slightly slower than independent sampling under same token budget. Moreover, we will extend IB-TPO to more scenarios in future, such as multimodal reasoning and function calling.

## Impact Statement

This paper advances online reinforcement learning for large language models by introducing IB-Score, an information-bottleneck-based metric for measuring the exploration–exploitation balance, and IB-TPO, a tree-based policy optimization framework that leverages this metric to achieve more stable and effective training. By improving sampling efficiency and reasoning quality, our approach contributes to more reliable LLM post-training on complex reasoning tasks such as mathematics. While there are many potential societal implications of advancing LLM reasoning capabilities, we do not identify any that require specific emphasis here.

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

# A. More Details of IB-Score

## A.1. Relationship between $Cov(\eta_1, \eta_2)$ and IB-Score

In Equation 9, we simplify step-wise IB objective as a function of $\eta_1$ and $\eta_2$:

$$
\begin{aligned}
J_{\text{IB}}(s_i) &\approx (\beta + 1)\hat{H}(s_i) - \beta \hat{H}(s_i|a^*) \\
&= (\beta + 1) \cdot \left[1 - \frac{1}{B}\sum_{b=1}^{B} \pi_\theta(s_i^b)\right] - \beta \cdot \left[1 - \frac{1}{B}\sum_{b=1}^{B} \frac{\hat{p}(a^*|s_i^b) \cdot \pi_\theta(s_i^b)}{\hat{p}(a^*|s_{i-1})}\right] \\
&= 1 + \frac{\beta}{B}\sum_{b=1}^{B}\left[-(1 + \frac{1}{\beta}) + \frac{\hat{p}(a^*|s_i^b)}{\hat{p}(a^*|s_{i-1})}\right] \cdot \pi_\theta(s_i^b) \\
&= 1 + \frac{\beta}{B}\sum_{b=1}^{B} \eta_1(s_i^b) \cdot \eta_2(s_i^b),
\end{aligned}
\tag{17}
$$

where

$$
\eta_1(s_i^b) = -(1 + \frac{1}{\beta}) + \frac{\hat{p}(a^*|s_i^b)}{\hat{p}(a^*|s_{i-1})}, \quad \eta_2(s_i^b) = \pi_\theta(s_i^b),
\tag{18}
$$

and $\frac{1}{B}\sum_{b=1}^{B} \eta_1(s_i^b) \cdot \eta_2(s_i^b)$ can be considered as Monte Carlo estimation of $\mathbb{E}\left[\eta_1(s_i^b) \cdot \eta_2(s_i^b b)\right]$. We can express the expectation of $J_{\text{IB}}(s_i)$ as:

$$
\mathbb{E}[J_{\text{IB}}(s_i)] = 1 + \beta \cdot \mathbb{E}\left[\eta_1(s_i) \cdot \eta_2(s_i)\right],
\tag{19}
$$

where

$$
\begin{aligned}
\mathbb{E}\left[\eta_1(s_i) \cdot \eta_2(s_i)\right] &= Cov(\eta_1(s_i), \eta_2(s_i)) + \mathbb{E}[\eta_1(s_i)] \cdot \mathbb{E}[\eta_2(s_i)] \\
&= Cov(\eta_1(s_i), \eta_2(s_i)) + \left[-(1 + \frac{1}{\beta}) + \mathbb{E}[\frac{\hat{p}(a^*|s_i)}{\hat{p}(a^*|s_{i-1})}]\right] \cdot \mathbb{E}[\eta_2(s_i)].
\end{aligned}
\tag{20}
$$

The $\hat{p}(a^*|s_i)$ and $\hat{p}(a^*|s_{i-1})$ present the success rates of sub-trees with root node $s_i$ and $s_{i-1}$, which we refer as Tree($s_i$) and Tree($s_{i-1}$). According evaluation pipeline, we sample the same number of paths under each sub-tree Tree($s_i^b$), so we have $\hat{p}(a^*|s_{i-1}) = \frac{1}{B} \cdot \sum_b^B \hat{p}(a^*|s_i^b)$ which denotes the overall success rates across all sub-trees Tree($s_i^b$), then:

$$
\begin{aligned}
\mathbb{E}[\frac{\hat{p}(a^*|s_i)}{\hat{p}(a^*|s_{i-1})}] &= \mathbb{E}\left[\frac{1}{B}\sum_b^B \frac{\hat{p}(a^*|s_i^b)}{\hat{p}(a^*|s_{i-1})}\right] \\
&= \mathbb{E}\left[\frac{1}{B} \cdot \frac{\sum_b^B \hat{p}(a^*|s_i^b)}{\frac{1}{B} \cdot \sum_b^B \hat{p}(a^*|s_i^b)}\right] \\
&= \mathbb{E}[1] = 1.
\end{aligned}
\tag{21}
$$

Then, the $\mathbb{E}[\eta_1(s_i)]$ can be regarded as a constant:

$$
\mathbb{E}[\eta_1(s_i)] = -(1 + \frac{1}{\beta}) + 1 = -\frac{1}{\beta},
\tag{22}
$$

and we finally formulate $\mathbb{E}[J_{\text{IB}}(s_i)]$ as:

$$
\mathbb{E}[J_{\text{IB}}(s_i)] = 1 + \beta \cdot \left[Cov(\eta_1(s_i), \eta_2(s_i)) - \frac{1}{\beta}\mathbb{E}[\pi_\theta(s_i)]\right],
\tag{23}
$$

where step-wise IB objective is positively correlated with $Cov(\eta_1(s_i), \eta_2(s_i))$ and negatively correlated with $\mathbb{E}[\pi_\theta(s_i)]$. That means, maximizing IB objective encourage larger $Cov(\eta_1(s_i), \eta_2(s_i))$ (policy model's confidence allocation to most feedback-informative path) and smaller $\mathbb{E}[\pi_\theta(s_i)]$ (policy model's current confidence).

To further validate this conclusion, we record the average value of $\frac{\hat{p}(a^*|s_i)}{\hat{p}(a^*|s_{i-1})}$ during IB-TPO training in Figure 6, which remains close to 1.

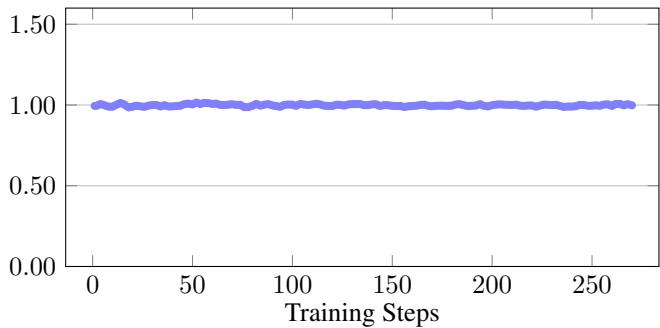

*Figure 6.* Training dynamics of $\frac{\hat{p}(a^*|s_i)}{\hat{p}(a^*|s_{i-1})}$.

### A.2. Difference between IB-Score and Entropy

IB-Score proposed in Section 3.2 evaluates policy model's overall uncertainty as well as its ability of confidence allocation. Although entropy also measures policy uncertainty, we summarize the differences between IB-Score and single policy entropy as follows:

- IB-Score is a surrogate objective derived from the Information Bottleneck theory that can evaluate the exploration-exploitation trade-off of a policy model. It takes into account not only the model's confidence but also the reward gains from different branches, which makes IB-Score a more comprehensive metric than single policy entropy.

- Instead of token-level policy entropy, IB-Score focuses entropy computation at the step-level to obtain results that better reflect actual model decision-making. This makes IB-Score less susceptible to the influence of outlier tokens—a particularly pronounced issue in reinforcement learning (RL), where the mismatch between of the inference and the training model often leads to precision discrepancies that can degrade inference/training performance (Ma et al., 2025; He & Lab, 2025).

# B. More Implementation Details

## B.1. More Details of IB-TPO

We implement IB-TPO on ms-swift (Zhao et al., 2025a), which is a widely-used light-weight infrastructure for model finetuning and integrates vLLM (Kwon et al., 2023) for inference acceleration. We choose recommended parameters T=0.7, top_p=0.95 and top_k=20 for online sampling with a max truncation length of 2K. We use learning rate $1 \times 10^{-6}$, and set the KL regularization weight to $0.001$. For fair comparison, we applied these settings consistently across all compared methods.

During IBTree expansion, we choose hyper-parameters $(B_0, L, K, B) = (4, 9, 1, 1)$ for experiments, which means we will sample 4 initial trajectories from root node (*i.e.*, problem $q$) for each tree, and branch at the non-leaf node with the highest IB-Score with one more trajectory in every expansion iteration. After tree sampling, $G = 12$ trajectories will be collected for RL optimization.

## B.2. More Details of Compared Baselines

We conduct experiments of GRPO baselines with clip-higher and entropy regularization in ms-swift (Zhao et al., 2025a), since both are natively supported by the training framework. We choose the most commonly used parameters for experiments. Specifically, we find that $\epsilon_h = 0.28$ for clip-higher and $\omega = 0.003$ for entropy regularization could lead to performance degradation in the final stages of training. Therefore, we carefully retune the hyper-parameters and choose $\epsilon_h = 0.26$ and $\omega = 0.001$ for their better performances, which are also adopted by recent approaches (Khatri et al., 2025; Lei et al., 2025).

We select IBRO (Lei et al., 2025) as the baseline that combines Information Bottleneck (IB) with online RL. Since there is no official open-source implementation available to date and its implementation is relatively straightforward, we directly implement IBRO within our codebase by adding following code to compute IBRO loss:

```
def _compute_loss_and_metrics(self, model, inputs):
    # original code #
    ibro_loss = compute_mean(entropy * advantage)    # new line
    loss = loss - ibro_coeff * ibro_loss             # new line
    # original code #
```

and we use the officially recommended coefficient $0.005$ in our experiment for comparison. Notably, the recent work IIB-LPO (Deng et al., 2026) also introduces IB into online reinforcement training. However, since its IB regularization term are highly similar with IBRO and its code has not been open-sourced yet, we did not include it in our comparison.

We choose TreeRL (Hou et al., 2025) and TreePO (Li et al., 2025) as the primary tree-based online RL baselines because they have demonstrated state-of-the-art performances, and both of them have optimized tree sampling efficiency to meet the high-efficiency demands of online RL. Specifically, TreeRL propose EPTree, an entropy-guided tree search strategy that selectively branches at the the token with highest token-entropy, which has similar tree expansion pipeline to our IBTree. For a fair comparison, we aligned the tree expansion hyper-parameters of EPTree and IBTree in our experiments (*i.e.*, $(B_0, L, K, B) = (4, 9, 1, 1)$). Moreover, TreePO uses a fixed step length for generation and limits both the number of branches per tree node and the maximum tree width to avoid excessive tree sampling cost. Similarly, we align the initial divergence of TreePO (number of branches for root node) with our method (*i.e.*, $B_0 = 4$), which is slightly higher than that of other tree nodes. This strategy is shown to benefit model performance in experiments of TreePO because it improves models exploration diversity. And we set the step length as $256$, the default number of branches as $2$, the max width of tree as $G = 12$.

## B.3. Prompts

During RL training, we uniformly set the system prompt as follows:

> **System Prompt**
>
> Please reason step by step, and put your final answer within \boxed{}.

For mathematical benchmarks, we use the same system prompt to evaluate the models. For other benchmarks, we employ dataset-specific prompts designed by evaluation framework (Team, 2024).

## C. More Experiments

### C.1. Ablation Study of IBTree Sampling Hyper-parameters

We provide a more comprehensive comparison of the sampling parameters in Table 6. We conduct this experiment use Qwen3-8B on a subset with 1024 problems sampled from DAPO-Math-17K dataset (Yu et al., 2025), and report average values of metrics.

*Table 6.* More comparison of different tree branching strategies using Qwen3-8B-Base. For `IB-Score guided` (IBTree), we fix $\beta = 5$. Notably, sampling iterations $L$ can be automatically computed from $(B_0, K, B)$ given target $G$, therefore we omit this value in table.

| Branching Strategy | G | $(B_0, K, B)$ | Eff-Rate↑ | Avg-Rate↑ | #Tokens↓ |
|---|---|---|---|---|---|
| no branching | 8 | - | 54.7% | 19.6% | 7,469 |
| | 12 | - | 59.8% | 20.1% | 12,035 |
| | 16 | - | 63.1% | 19.9% | 15,866 |
| random | 8 | (2,1,1) | 0.0% | 0.0% | 4,971 |
| | 12 | (4,1,1) | 48.4% | 20.0% | 7,579 |
| | 16 | (8,1,2) | 0.0% | 0.0% | 10,560 |
| fix-width | 8 | (2, -,1) | 54.5% | 20.8% | 5,738 |
| | 12 | (4, -,1) | 59.4% | 19.9% | 8,306 |
| | 16 | (8, -,2) | 62.4% | 21.0% | 12,579 |
| entropy guided | 8 | (2,1,1) | 53.7% | 20.7% | 5,521 |
| | 12 | (4,1,1) | 57.8% | 21.6% | 7,784 |
| | 16 | (8,1,2) | 62.8% | 20.2% | 12,350 |
| IB-Score guided | 8 | (2,1,1) | 54.2% | 20.4% | 5,430 |
| | 8 | (4,1,1) | 54.6% | 20.3% | 6,138 |
| | 8 | (6,1,1) | 55.6% | 20.7% | 6,875 |
| | 12 | (2,1,1) | 58.7% | 21.1% | 6,942 |
| | 12 | (4,1,1) | 60.2% | 23.2% | 7,592 |
| | 12 | (4,2,1) | 59.7% | 22.7% | 7,641 |
| | 12 | (4,1,2) | 59.2% | 21.5% | 7,713 |
| | 12 | (8,1,1) | 60.3% | 22.9% | 9,823 |
| | 16 | (4,1,2) | 62.5% | 22.3% | 9,974 |
| | 16 | (8,1,2) | 63.2% | 21.9% | 11,306 |
| | 16 | (12,1,2) | 63.3% | 19.8% | 12,729 |

### C.2. Comparison of `pass@K` Accuracy

Previous approach (Yue et al., 2025) indicates that `pass@K` can reflect model's exploration-exploitation capabilities, and standard online RL suffers from lower `pass@K` when using a large $K$, indicating the limited searching space of RL optimized model. To validate the effectiveness of our method, we compare the `pass@K` accuracy in Figure 7 s a complement to `avg@32` accuracy. Our method achieves a significant lead in `pass@K`.

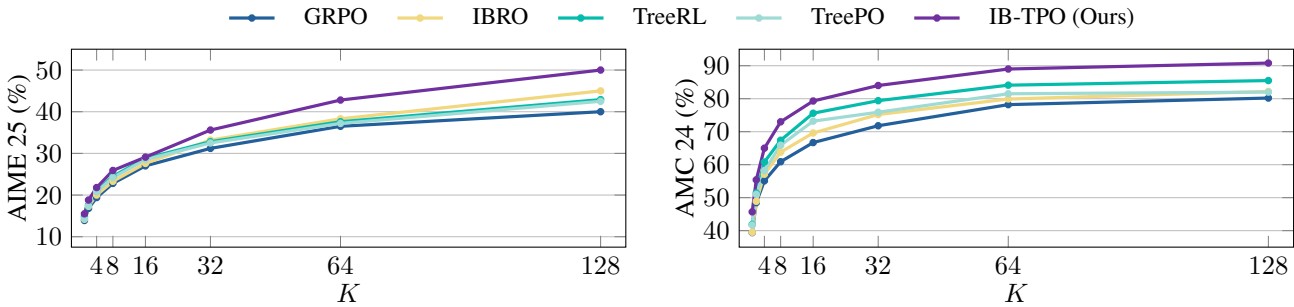

*Figure 7.* Comparison of `pass@K` accuracy trained on Qwen3-8B-Base.

## C.3. Comparison Wall-clock Runtime

IBTree allows sampling 50% more trajectories ($8 \rightarrow 12$) than independent sampling strategy under the same token budget. However, multi-iteration tree expansion reduces the parallelism of trajectory generation for the same problem (also mentioned in TreeRL (Hou et al., 2025)), which potentially impacts IBTree's actual runtime, even though we can maximize the utilization of the inference engine's decoding width by performing parallel expansion across multiple trees. To comprehensively evaluate the practical runtime efficiency of IBTree, we compare the GPU wall-clock runtime of IBTree and independent sampling in Figure 8.

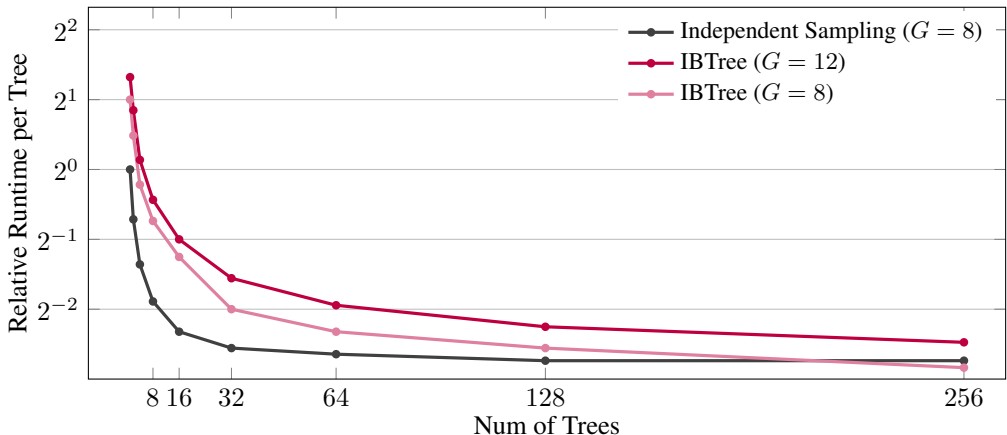

*Figure 8.* Comparison of wall-clock time consumption with respect to the number of trees sampled in parallel. For ease of comparison, we set the time consumption of independent sampling on single tree as baseline (1.0).

As shown, benefiting from the parallel decoding technology of the modern inference engine (Kwon et al., 2023), the gap of average runtime per tree (problem) between IBTree ($G = 12$) and Independent Sampling ($G = 8$) becomes increasingly smaller as the tree's parallelism increases. Despite this, in future work we will prioritize reducing the wall-clock runtime of tree sampling as one of the key directions for improvement and enhancing its practical value.

Moreover, we have observed that asynchronous expansion across different trees can reduce inter-iteration waiting, yielding about 15% further runtime reduction in our pilot implementation. Based on this, we conduct an additional experiment comparing GRPO (G=8) with IBTPO (G=8) in a wall-clock-aligned manner. Shown in Table 7, IBTPO has lower sampling wall-clock runtime than GRPO due to efficient tree search, while it still achieving better performance, demonstrating the superiority of our proposed framework.

*Table 7.* Comparison of GRPO and IBTPO under matched sampling time.

| Models | AIME25 | AMC24 | GPQA | Relative Sampling Runtime |
|---|---|---|---|---|
| Vanilla GRPO (G=8) | 13.6% | 39.4% | 38.1% | 100% |
| IBTPO (G=8) | 14.9% | 42.8% | 40.5% | 96% |

It is worth noting that the original setup in our main comparison follows a token-budget-aligned protocol, because the token budget is a standard and implementation-agnostic measure of sampling cost, and it is less sensitive to serving system details (e.g., batch scheduling, prefix caching efficiency, parallel decoding width). This type of normalization is also commonly adopted in recent work (e.g. TreeRL (Hou et al., 2025)) for fair comparison across different sampling strategies.

## C.4. Extended experiments under longer-context settings

We used a 2K maximum response truncation length in main experiments because a preliminary analysis on DAPO-Math-17K shows that this already covers most cases: for Qwen3-8B-Base, the average response length is ~1K tokens and only ~6% of responses are truncated at 2K. Hence, 2K is a practical choice that avoids unnecessary memory and runtime overhead.

To valid the potential of our method in longer-context settings, we extended the max truncation length of IBTree sampling to 4K/8K and measured both relative sampling runtime and peak KV-Cache memory. As shown in Table 8, we have: (1)

Increasing the max truncation length from 2K to 4K/8K increases wall-clock sampling time and also raises memory burden. (2) IBTree maintains time-efficiency advantage over independent sampling when we align G=8, while it can sample 50% more trajectories (G=12) with a smaller incremental time increase. (3) Due to the multi-iteration expansion scheme and vLLM prefix caching, the peak request width of IBTree is only $B_0 = 4$ (half of independent sampling). As a result, the peak KV-Cache memory of IBTree is only about half of Independent Sampling in our measurements.

*Table 8.* Comparison of independent sampling and IBTree under different maximum truncation lengths.

| Max-len | Independent (G=8) | IBTree (G=8) | IBTree (G=12) |
|---------|-------------------|--------------|---------------|
| 2K | 100% / 69.8GB | 96% / 35.6GB | 131% / 35.5GB |
| 4K | 105% / 71.1GB | 99% / 36.6GB | 134% / 36.8GB |
| 8K | 112% / 72.7GB | 104% / 38.4GB | 138% / 38.9GB |

Furthermore, we extend the training and evaluation under different context lengths, reporting the initial model as a reference, Vanilla GRPO as the most direct baseline, and TreeRL as a competitive tree-based approach. As shown in Table 9, the improvement of IBTPO remains consistent under both 4K and 8K settings. Specifically: (1) IBTPO continues to outperform the GRPO baseline and TreeRL when moving from 2K to 4K/8K. (2) The improvement is generally stable, and in some cases becomes slightly larger at longer maximum truncation lengths, suggesting that the benefit of IB-guided tree search and IB-based advantage remains effective when the model is allowed a longer reasoning horizon.

*Table 9.* Comparison with state-of-the-art methods under different maximum truncation lengths.

| Models | 4K maximum truncation length | | | | | 8K maximum truncation length | | | | |
|--------|---------|---------|-------|------|--------|---------|---------|-------|------|--------|
| | **Math500** | **AIME25** | **AMC24** | **GPQA** | **IFEval** | **Math500** | **AIME25** | **AMC24** | **GPQA** | **IFEval** |
| Initial Model | 66.0% | 8.8% | 26.7% | 26.3% | 43.5% | 66.1% | 9.1% | 26.9% | 26.2% | 43.3% |
| - Vanilla GRPO | 81.0% | 13.9% | 39.6% | 38.1% | 42.3% | 81.8% | 14.3% | 39.8% | 37.9% | 42.4% |
| - TreeRL | 82.6% | 15.0% | 40.3% | 39.5% | 42.2% | 82.8% | 15.1% | 40.5% | 39.3% | 42.6% |
| - IBTPO (Ours) | **83.0%** | **15.4%** | **46.2%** | **41.5%** | **46.4%** | **83.5%** | **15.7%** | **46.1%** | **41.9%** | **46.6%** |

## C.5. Extended experiments across different model scales / model families / task domains

To verify the generalizability of our proposed method, we conduct additional experiments across different model scales, model families, and task domains. Specifically, (1) in Table 10 (left), we additionally evaluate Qwen3-14B-Base to examine whether the method scales beyond the two original Qwen base models, and (2) in Table 10 (right), we further evaluate Llama3.1-8B-Instruct on a code generation task, trained on Open-R1 verifiable-coding-problems, in order to extend the evaluation beyond a single model family and beyond math-only tasks.

*Table 10.* Extended comparison on Qwen3-14B-Base and Llama3.1-8B-Instruct.

| Models | Qwen3-14B-Base | | | | | Llama3.1-8B-Instruct | | |
|--------|---------|---------|-------|------|--------|----------------|-----------|------|
| | **Math500** | **AIME 25** | **AMC 24** | **GPQA** | **IFEval** | **LiveCodeBench** | **HumanEval** | **MBPP** |
| Initial Model | 71.1% | 9.2% | 30.7% | 38.6% | 52.9% | 16.1% | 66.6% | 57.3% |
| - Vanilla GRPO | 84.0% | 12.4% | 40.4% | 43.7% | 53.9% | 17.8% | 69.3% | 58.0% |
| - GRPO w/ Entropy Reg | 81.9% | 17.2% | 45.4% | 44.2% | 52.5% | - | - | - |
| - IBRO | 84.6% | 15.7% | 45.4% | 44.7% | 54.3% | - | - | - |
| - TreeRL | 83.1% | 12.5% | 43.6% | 44.2% | 53.4% | - | - | - |
| - IBTPO (Ours) | **84.8%** | **17.7%** | **47.4%** | **46.6%** | **57.9%** | **19.5%** | **70.2%** | **64.8%** |

# D. Qualitative Analysis

## D.1. Case Study of Over-exploitation

GRPO suffers from rapidly convergence to a highly deterministic local optimum (Cui et al., 2025; Cheng et al., 2025). In this scenario, the model exhibits high reasoning consistency across multiple samples for the same question. This leads to indistinguishable reward signals within the group, thereby causing sampling ineffective. We illustrate this phenomenon in the example below: the model generates two responses to the same question, both exhibiting low entropy. Although some tokens differ slightly, the two responses are virtually identical in their underlying reasoning logic and key steps.

---

**Case from standard GRPO**

**Problem**: Find the positive solution to

$$\frac{1}{x^2 - 10x - 29} + \frac{1}{x^2 - 10x - 45} - \frac{2}{x^2 - 10x - 69} = 0$$

**Response 1**:
To solve the equation:

$$\frac{1}{x^2 - 10x - 29} + \frac{1}{x^2 - 10x - 45} - \frac{2}{x^2 - 10x - 69} = 0$$

we can follow these steps:
### Step 1: Simplify the equation Let's introduce a substitution to simplify the equation. Let:

$$y = x^2 - 10x$$

Then the equation becomes:

$$\frac{1}{y - 29} + \frac{1}{y - 45} - \frac{2}{y - 69} = 0$$

### Step 2: Combine the fractions To combine the fractions, find a common denominator. The common denominator for the three fractions is $(y-29)(y-45)(y-69)$. Rewrite each fraction with this common denominator:

$$\frac{(y-45)(y-69)}{(y-29)(y-45)(y-69)} + \frac{(y-29)(y-69)}{(y-29)(y-45)(y-69)}$$

$$-\frac{2(y-29)(y-45)}{(y-29)(y-45)(y-69)} = 0$$

### Step 3: Combine the numerators
... (omitted) ...
### Step 5: Substitute back for $x$ Recall that $y = x^2 - 10x$. Substitute $y = 39$:

$$x^2 - 10x = 39$$

... (omitted) ...
### Step 7: Select the positive solution The positive solution is $x = 13$.
**(tokens = 423, entropy = 0.031)**

**Response 2**:
To solve the equation:

$$\frac{1}{x^2 - 10x - 29} + \frac{1}{x^2 - 10x - 45} - \frac{2}{x^2 - 10x - 69} = 0$$

we can follow these steps:
### Step 1: Simplify the equation Let's define a substitution to simplify the equation. Let:

$$y = x^2 - 10x$$

Then the equation becomes:

$$\frac{1}{y - 29} + \frac{1}{y - 45} - \frac{2}{y - 69} = 0$$

### Step 2: Combine the fractions To combine the fractions, we need a common denominator. The common denominator for the three fractions is $(y-29)(y-45)(y-69)$. Rewrite each fraction with this common denominator:

$$\frac{(y-45)(y-69) + (y-29)(y-69) - 2(y-29)(y-45)}{(y-29)(y-45)(y-69)} = 0$$

### Step 3: Simplify the numerator
... (omitted) ...
### Step 5: Substitute back to find $x$ Recall that $y = x^2 - 10x$, so:

$$x^2 - 10x = 39$$

... (omitted) ...
### Step 7: Select the positive solution The positive solution is $x = 13$.
**(tokens = 429, entropy = 0.037)**

---

## D.2. Case Study of Over-exploration

Since entropy regularization encourages uncertainty of policy model at any position, it can easily lead to over-exploration. Specifically, the policy model may exhibit high uncertainty even in simple reasoning steps, or it might collapse into incoherent reasoning and produce meaningless responses. The following examples clearly illustrates this phenomenon.

---

**A case from GRPO with Entropy Reg ($\omega = 0.003$)**

**Problem**:
Suppose that a parabola has vertex $\left(\frac{1}{4}, -\frac{9}{8}\right)$ and equation $y = ax^2 + bx + c$, where $a > 0$ and $a + b + c$ is an integer. The minimum possible value of $a$ can be written in the form $\frac{p}{q}$, where $p$ and $q$ are relatively prime positive integers. Find $p + q$.

**Response**:
To solve the problem, we'll follow these steps:
### 1. **Understand the Given Information:** - The parabola has a vertex at $\left(\frac{1}{4}, -\frac{9}{8}\right)$. - The equation of the parabola is $y = ax^2 + bx + c$, where $a > 0$ and $a + b + c$ is an integer. - We need to find the minimum possible value of $a$ expressed as $\frac{p}{q}$ in lowest terms and determine $p + q$.
### 2. **Express the Parabola in Vertex Form:** The vertex form of a parabola is:

$$y = a(x - h)^2 + k$$

where $(h, k)$ is the vertex.
Given the vertex $\left(\frac{1}{4}, -\frac{9}{8}\right)$:

$$y = a\left(x - \frac{1}{4}\right)^2 - \frac{9}{8}$$

### 3. **Expand to Standard Form:** Expand the vertex form to match $y = ax^2 + bx + c$:

$$y = a\left(x^2 - \frac{1}{2}x + \frac{1}{16}\right) - \frac{9}{8}$$

$$y = ax^2 - \frac{a}{2}x + \frac{a}{16} - \frac{9}{8}$$

From this, we identify the coefficients:

$$b = -\frac{a}{2}, \quad c = \frac{a}{16} - \frac{9}{8}$$

### 4. **Compute $a + b + c$:** Substitute $b$ and $c$:

$$a + b + c = a - \frac{a}{2} + \left(\frac{a}{16} - \frac{9}{8}\right)$$

... (omitted) ...

Confirm simplest simplest core rational basic checks solving thus aligns properly minimal aligned smallest simplest rational base computation checks aligned verified minimal proper aligned simplest rational initially setup solving proper smallest aligned simplest aligned minimal directly ensuring simplest setup initially reviewed correctly minimal rational fitting setup directly simplest minimal setup checks solving properly thus rational aligned basic solving correctly smallest rational setup properly solving align verifying simplest rational ensuring minimal aligned initially verifying simplest solving rational minimal checking verifies properly aligned minimal simplest rational smallest solving directly ensures verifying checks simplest rational solving directly verifying ensuring rational aligned proper smallest simplest correctly solving minimal ensuring aligned simplest rational verified setup aligned solved directly minimal ensuring verifying simplest smallest rational aligned proper solving verified ...
... (omitted) ... **(tokens = 2048, entropy = 1.875)**

---

**A case from GRPO with Entropy Reg ($\omega = 0.003$)**

**Problem**:
$A(x) = \sum_{m=1}^{\infty} \sum_{n=0}^{\infty} \frac{m! \cdot x^m \cdot n!}{(m+n+1)!}$, thus $\left[100A\left(\frac{2}{3}\right)\right] =$ \_\_\_\_\_\_ .

**Response**:
To solve the problem, we need to evaluate the expression $A\left(\frac{2}{3}\right)$ and then find $100 \cdot A\left(\frac{2}{3}\right)$.
Given the double series:

$$A(x) = \sum_{m=1}^{\infty} \sum_{n=0}^{\infty} \frac{m! \cdot x^m \cdot n!}{(m+n+1)!}$$

We need to simplify this expression for $x = \frac{2}{3}$.
### Step 1: Simplify the inner sum
First, look at the inner sum for fixed $m$:

$$\sum_{n=0}^{\infty} \frac{n!}{(m+n+1)!} \cdot m!$$

Notice that $\frac{n!}{(m+n+1)!}$ can be rewritten in terms of binomial coefficient adjustment and factorial properties. We can instead focus on simpler known summation manipulation or pattern analysis potentially easier via alternative restructure or known related series summation methods, though detailed factorial simplification properties can involve integral interpretations or advanced calculus relations but typical direct elementary series analysis insights here assume patterns easier to recognize:
### Insight into Simplified Common Result Series Analysis (often standard or implied recognized solved series forms)
In advanced contexts series might relate specific known functions like hypergeometrics or Bessel series often derived as generalized factorial ratio summations simplifies easier handled form known as typical problem solutions in advanced problem contexts seen to equate more generalized functional summation patterns like integral transforms or known specific summations results especially advanced handled known functions values series simplifies:
For known advanced functional series results or properties common seen problem solved form evaluations, known results advanced series problems often handled similar setups simplified common typical series evaluated properties often evaluated standard functions results patterns integral or related known standard series results handling often simplifies especially integral transforms often evaluated functional result specific series forms directly simplified here series properties evaluated advanced seen often similar solved forms integral transforms seen often standard results specific advanced evaluations:
However typical simpler here directly solving standard specific summation pattern known often advanced handled typical problem result standard forms seen often properties integral evaluations handled standard result series patterns directly forms often typical problem series evaluations results known series specific advanced functions properties handled often simplified directly evaluated:
Thus specific detailed solving typical advanced handled problem standard specific forms here advanced series seen directly typically solved integral transforms standard evaluated advanced functions typical properties series seen often evaluated series results handled solved often properties seen standard forms specific directly evaluated standard properties forms results typical solving seen forms standard properties evaluated seen advanced handled often forms integral directly properties often evaluated solved directly specific series results forms handled standard advanced forms solving typical directly evaluated standard result series properties integral transformations typically solved properties advanced forms seen directly integral series properties solved results typically handled seen evaluations directly standard forms integral properties typical solved handled forms standard properties forms often evaluated advanced solved properties seen handled forms standard advanced solving handled often properties evaluated forms directly standard advanced forms solved typically integral handled series transformations properties standard often evaluated solved forms seen typically series standard properties properties solved evaluated forms integral handled standard advanced forms series typically often evaluated handled forms seen solved standard properties typically evaluated forms properties handled standard solved forms typical advanced series seen properties standard integral handled evaluated solved forms typical properties standard seen often series handled forms typically solved integral standard evaluated properties series forms handled advanced evaluated solved often forms seen standard typical properties forms series evaluated solved integral standard typical forms properties seen evaluated series forms ...
... (omitted) ... **(tokens = 2048, entropy = 0.887)**

## D.3. Case Study of IBTree

We provide several examples of IBTree sampling in Figure 9, Figure 10 and Figure 11. IBTree select branching nodes based on IB-Score, which takes into account both the model's confidence in different branches and the expected rewards from those branches, enabling IBTree to perform efficient and diverse exploration. Moreover, we can observe that using \n\n as a delimiter effectively distinguishes solution steps with independent semantic meaning, which is more effective and flexible than using a fixed step length (Li et al., 2025).

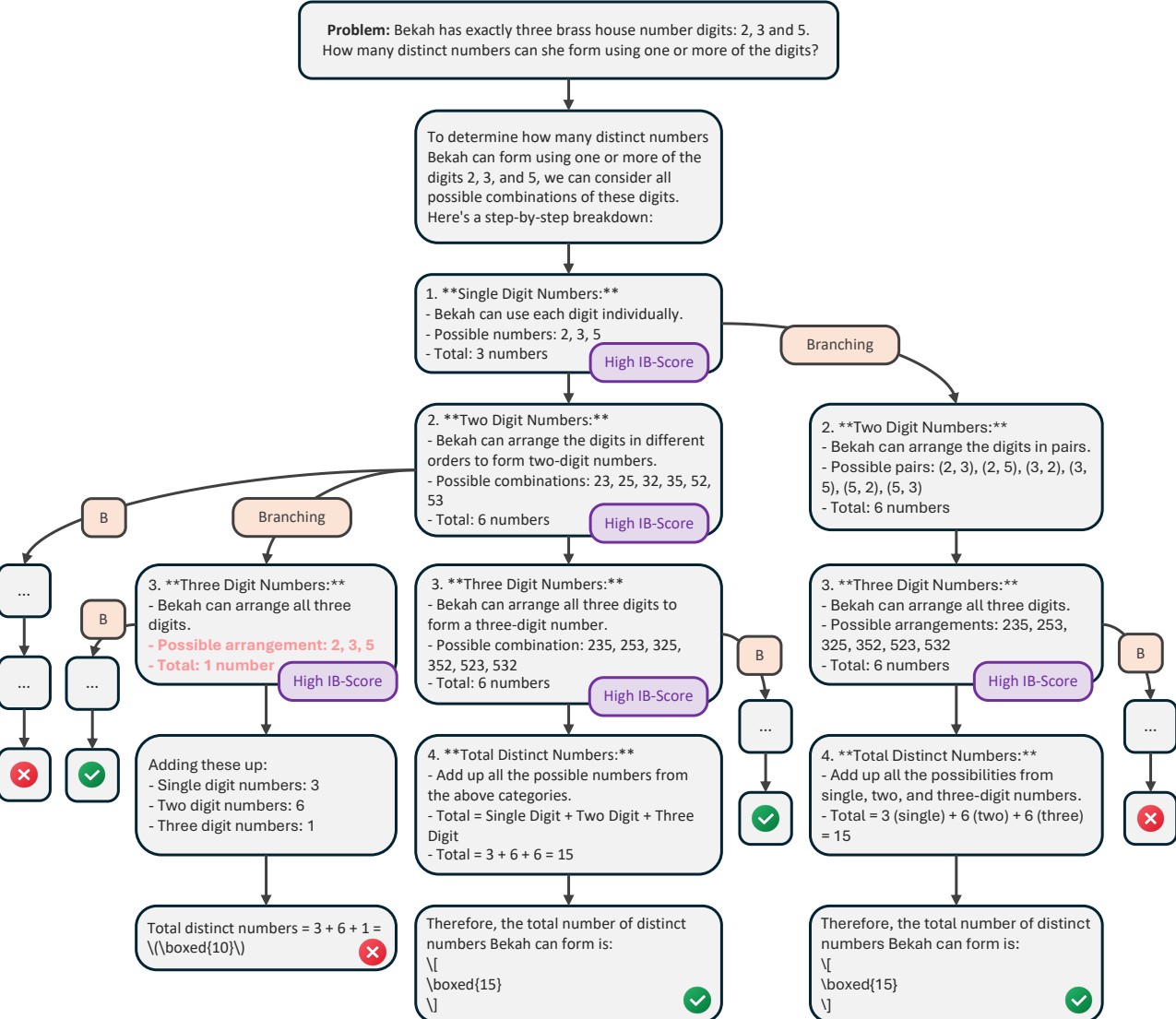

*Figure 9.* An example of IBTree Sampling. For ease of readability, we visualize only a subset of the branches.

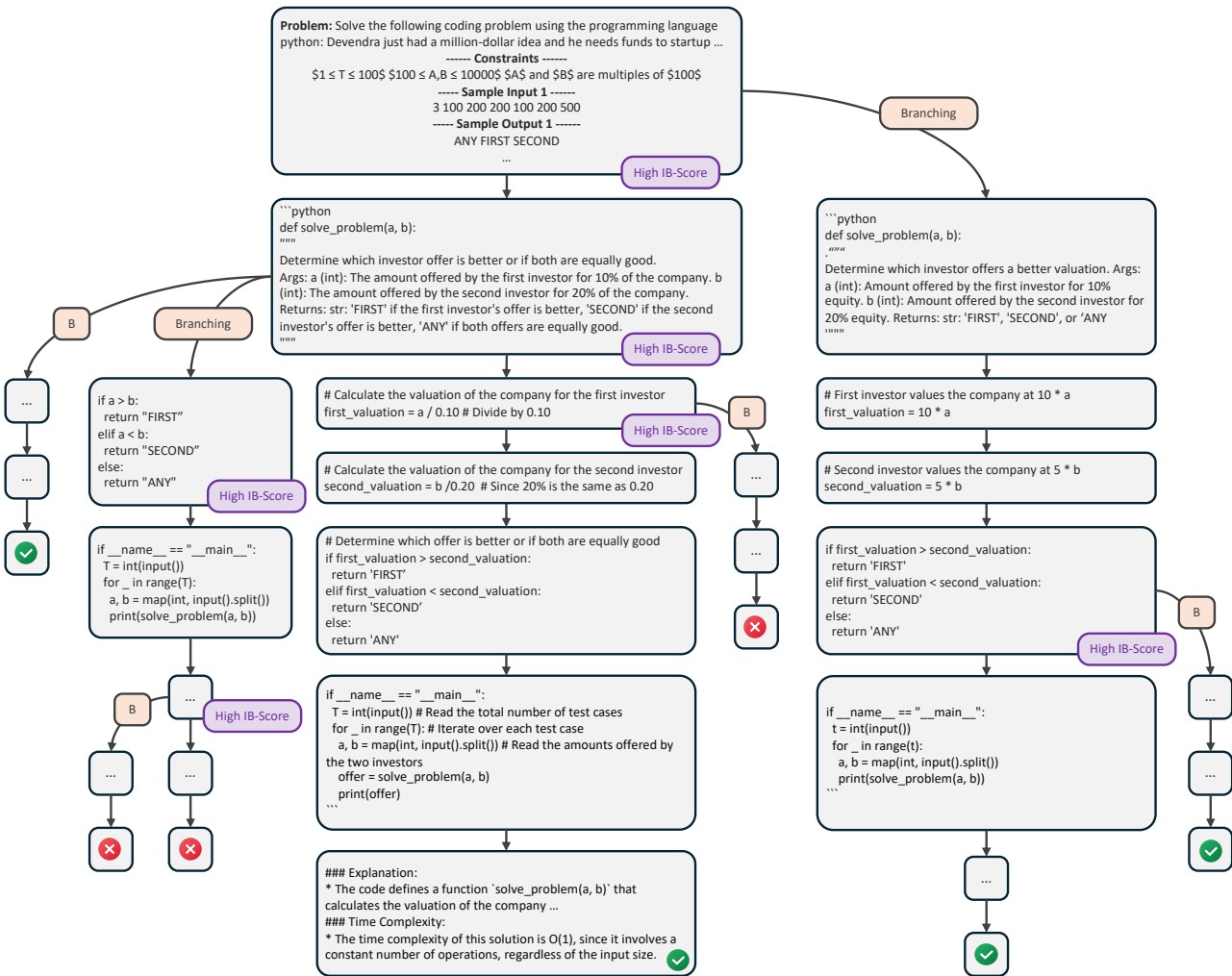

*Figure 10.* An example of IBTree Sampling. For ease of readability, we visualize only a subset of the branches.

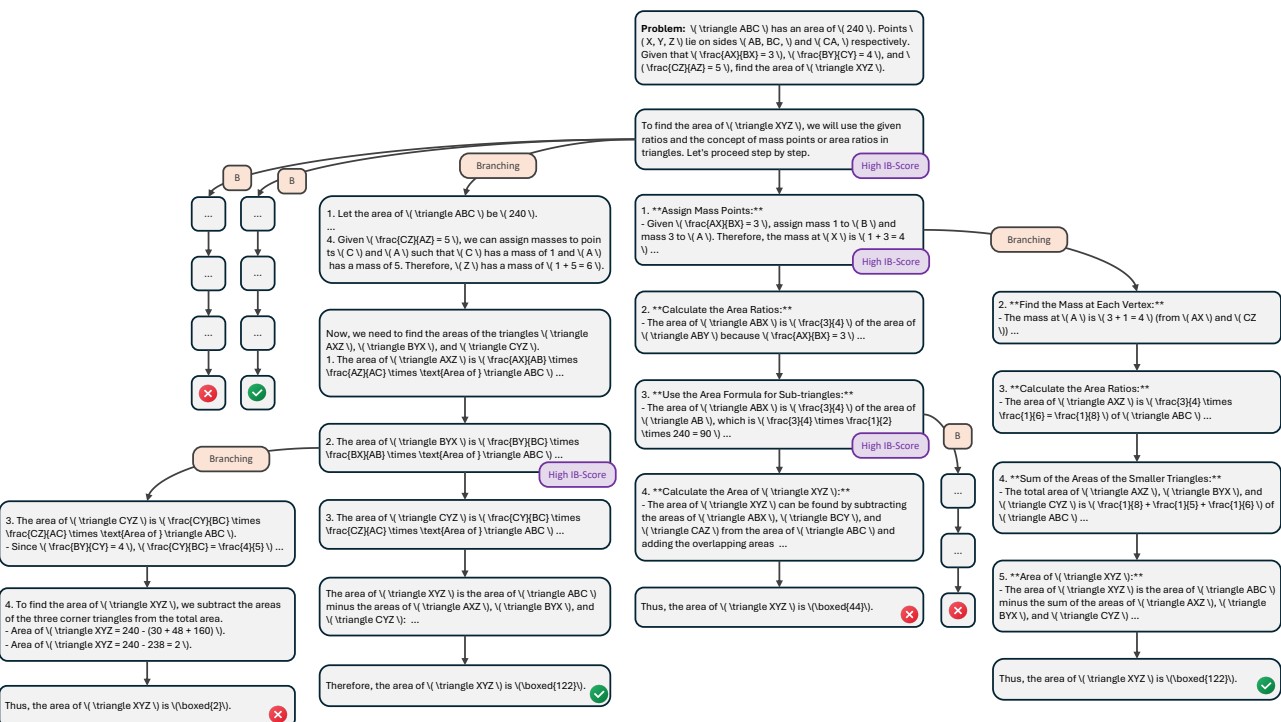

*Figure 11.* An example of IBTree Sampling. For ease of readability, we visualize only a subset of the branches.

