# OpenReview forum: "Long Live The Balance: Information Bottleneck Driven Tree-based Policy Optimization"
_ICML.cc/2026/Conference — ICML 2026 regular_

### Official Review · Reviewer_uzxw · 2026-02-23

**Soundness:** 3
**Presentation:** 3
**Significance:** 3
**Originality:** 3
**Overall Recommendation:** 4
**Confidence:** 4

**Summary:**

This paper studies exploration-exploitation imbalance in RL for LLM reasoning. The paper proposes IB-Score, a novel metric grounded in Information Bottleneck that evaluates policy’s exploration-exploitation balance by quantifying the tradeoff between step-level reasoning diversity and mutual information shared with the correct answer. Based on the IB-Score, the paper proposes Information Bottleneck-driven Tree-based Policy Optimization (IB-TPO), which uses IB-Score to guide tree sampling. Experiments on Qwen3 1.7B and 8B models show consistent improvements over baselines.

**Compliance With Llm Reviewing Policy:**

Affirmed.

**Final Justification:**

The rebuttal addressed my concerns. I raise my score to weak accept.

**Key Questions For Authors:**

Please see weaknesses

**Limitations:**

yes

**Strengths And Weaknesses:**

Strengths

1.	The main results show consistent gains compared to prior methods.
2.	The ablations are detailed.

Weaknesses

1.	The experiment setting seems not very fair for the GRPO baseline. From Appendix C.3, the sampling time for tree-based method are higher than that our GRPO. But the training setup allows tree-based methods to sample more trajectories (12) than GRPO (8). I would like to see the result of GRPO by matching the wall-clock time in sampling.
2.	The training setup uses a maximum sequence length of 2K tokens, which is relatively short. It is unclear whether the proposed multi-iteration tree expansion remains practical or beneficial at larger context lengths like 4K and 8K, where the parallelism and benefit of tree expansion would be reduced. I would like to see scalability experiments reporting wall-clock time and memory usage under longer max-length settings.
3.	The paper uses \n\n to delimit steps. Is this robust or model-dependent. I would like to see analysis of this design.

---

> ### Author Rebuttal · Authors · 2026-03-30
>
> We sincerely thank the Reviewer `uzxw` for the careful reading and for highlighting three important aspects: (W1) fairness of the GRPO comparison under wall-clock constraints, (W2) scalability to longer context lengths, and (W3) the robustness of using \n\n as the step delimiter. We appreciate these suggestions and have conducted additional analyses/experiments accordingly. We provide our point-by-point responses below.
>
> > **W1: The experiment setting seems not very fair for the GRPO baseline ... I would like to see the result of GRPO by matching the wall-clock time in sampling.**
>
> Thank you for this important suggestion. We agree that wall-clock-aligned comparison is an important complementary perspective to token-budget-aligned comparison.
>
> Our original setup follows a token-budget-aligned protocol, because token budget is a standard and implementation-agnostic measure of sampling cost, and it is less sensitive to serving system details (e.g., batch scheduling, prefix caching efficiency, parallel decoding width). This type of normalization is also commonly adopted in recent work (e.g. TreeRL) for fair comparison across different sampling strategies.
>
> We also agree that the current multi-iteration tree expansion can be further optimized. In fact, we have observed that **asynchronous expansion** across different trees can further reduce inter-iteration waiting, yielding about **15% runtime reduction** in our pilot implementation.
>
> Following your suggestion, we conducted an additional experiment comparing GRPO (G=8) with IBTPO (G=8) in a wall-clock-aligned manner:
>
> |Models|AIME25|AMC24|GPQA|Relative Sampling Runtime|
> |-|-|-|-|-|
> |Vanilla GRPO (G=8)|13.6%|39.4%|38.1%|100%|
> |IBTPO (G=8)|**14.9%**|**42.8%**|**40.5%**|**96%** (optimized)|
>
> where IBTPO has lower sampling wall-clock time than GRPO due to efficent tree search, while still achieving better performance. We will include this optimization and comparison in the revised paper.
>
> > **W2: The training setup uses a maximum sequence length of 2K tokens, which is relatively short ... I would like to see scalability experiments reporting wall-clock time and memory usage under longer max-length settings.**
>
> Thank you for this important suggestion. We used a 2K maximum response truncation length in main experiments because a preliminary analysis on DAPO-Math-17K shows that this already covers most cases: for Qwen3-8B-Base, **the average response length is ~1K tokens and only ~6% of responses are truncated at 2K**. Hence, 2K is a practical choice that avoids unnecessary memory and runtime overhead.
>
> Following your suggestion, we extended the max truncation length to 4K/8K and measured both ***relative sampling runtime** and **peak KV-Cache memory***:
> |Max-len|Independent(G=8)|IBTree(G=8)|IBTree(G=12)|
> |-|-|-|-|
> |2K|100% / 69.8GB|96% / 35.6GB|131% / 35.5GB|
> |4K|105% / 71.1GB|99% / 36.6GB|134% / 36.8GB|
> |8K|112% / 72.7GB|104% / 38.4GB|138% / 38.9GB|
>
> The results show that: (1) Increasing the max truncation length from 2K to 4K/8K increases wall-clock sampling time and also raises memory burden. (2) IBTree maintains time-efficiency advantage over independent sampling when we align G=8, while it can sample 50% more trajectories (G=12) with a smaller incremental time increase. (3) Due to the multi-iteration expansion scheme and vLLM prefix caching, the peak request width of IBTree is only $B_0 = 4$ (half of independent sampling). As a result, the peak KV-Cache memory of IBTree is only about half of Independent Sampling in our measurements.
>
> We will include these scalability results in the revised paper.
>
> > **W3: The paper uses \n\n to delimit steps. Is this robust or model-dependent. I would like to see analysis of this design.**
>
> We agree that the step delimiter is an implementation detail worth clarifying. We use \n\n because it is a simple, training-free, and natural formatting boundary in chain-of-thought responses. Importantly, in our framework \n\n is mainly used to provide candidate branching points in the partial trajectory, rather than to define a strict semantic decomposition that must be perfectly correct. We would also like to emphasize that the current experiments (in Table 1 and our additional experiments for Q1 of the Reviewer `LWZc`) provide direct evidence that this design generalizes beyond a single setting, where IB-TPO consistently works across different model families / model scales / task domains.
>
> To directly test robustness, we additionally ran a noise-injection experiment on Qwen3-8B-Base: we randomly replace 10% of existing \n\n split points with random selected positions to simulate under-segmentation and over-segmentation noise:
>
> | Models | AIME25 | AMC24 | GPQA |
> | -| - | - | - |
> | w/o noise | 15.3% | 46.0% | 41.7% |
> | w/ noise | 15.5% | 45.7% | 41.5% |
>
> The performance change was not significant, indicating that IB-TPO is robust to moderate step-boundary perturbations. We will include these results and discussion in the revised paper.

---

> > ### Author Rebuttal · Reviewer_uzxw · 2026-04-01
> >
> > The rebuttal addresses most of my concerns. I have one remaining question about the added experiments with maximum sequence lengths of 4K and 8K. Could you report the benchmark performance of IBTPO alongside a few baselines under these settings? Does the improvement remains or becomes higher with higher maximum lengths?

---

> > > ### Author Response · Authors · 2026-04-04
> > >
> > > We thank the reviewer `uzxw` for the follow-up question and for acknowledging that our rebuttal addressed most of the earlier concerns.
> > >
> > > To directly address the reviewer’s question on whether the gains persist under higher maximum truncation lengths, we additionally evaluate Qwen3-8B-Base under 4K and 8K maximum truncation lengths. We report the initial model as a reference, Vanilla GRPO as the most direct baseline, and TreeRL as a competitive tree-based approach. The results are as follows:
> > >
> > > > **4K maximum truncation length**
> > >
> > > | Models | Math500 | AIME25 | AMC24 | GPQA | IFEval |
> > > |-|-|-|-|-|-|
> > > | Initial Model | 66.0% | 8.8% | 26.7% | 26.3% | 43.5% |
> > > | - Vanilla GRPO | 81.0% | 13.9% | 39.6% | 38.1% | 42.3% |
> > > | - TreeRL | 82.6% | 15.0% | 40.3% | 39.5% | 42.2% |
> > > | - IBTPO (Ours) | **83.0%** | **15.4%** | **46.2%** | **41.5%** | **46.4%** |
> > >
> > > > **8K maximum truncation length**
> > >
> > > | Models | Math500 | AIME25 | AMC24 | GPQA | IFEval |
> > > |-|-|-|-|-|-|
> > > | Initial Model | 66.1% | 9.1% | 26.9% | 26.2% | 43.3% |
> > > | - Vanilla GRPO | 81.8% | 14.3% | 39.8% | 37.9% | 42.4% |
> > > | - TreeRL | 82.8% | 15.1% | 40.5% | 39.3% | 42.6% |
> > > | - IBTPO (Ours) | **83.5%** | **15.7%** | **46.1%** | **41.9%** | **46.6%** |
> > >
> > > **Overall, the improvement of IBTPO remains consistent under both 4K and 8K settings.**
> > >
> > > Specifically:
> > >
> > > 1. IBTPO continues to outperform the GRPO baseline and TreeRL when moving from 2K to 4K/8K.
> > > 2. The improvement is generally stable, and in some cases becomes slightly larger at longer maximum truncation lengths, suggesting that the benefit of IB-guided tree search and IB-based advantage remains effective when the model is allowed a longer reasoning horizon.
> > >
> > > We will include these results and discussion in the revised paper for completeness.

---

### Official Review · Reviewer_LWZc · 2026-03-03

**Soundness:** 3
**Presentation:** 3
**Significance:** 3
**Originality:** 3
**Overall Recommendation:** 5
**Confidence:** 4

**Summary:**

In this paper, the authors propose IB-TPO, a method for addressing the exploration–exploitation trade-off in RL for LLMs (RL4LLMs). The paper is clearly written and easy to follow. From my perspective, the core idea is novel, and the experimental results indicate that the proposed method is effective.
Although the approach introduces several additional hyperparameters that require tuning, and the experimental evaluation is limited to MATH-related datasets. Overall, I consider this a strong paper.

**Compliance With Llm Reviewing Policy:**

Affirmed.

**Final Justification:**

My concerns have been largely resolved during rebuttal, and I have raised my score and confidence accordingly.

**Key Questions For Authors:**

1. Although the paper provides detailed ablation studies for each hyperparameter, it remains unclear whether the recommended settings would transfer to other base models or tasks (coding/agent). I would appreciate discussion or evidence on how sensitive these hyperparameters are to changes in model family, model scale, or task domain.

**Limitations:**

yes

**Strengths And Weaknesses:**

### Strength
1. In my view, the idea is novel and may be the first to introduce an IB perspective to address the exploration–exploitation trade-off in RL for LLMs.
2. The paper is well-written and easy to follow and I enjoy the reading.
3. The theoretical explanation is convincing, and the ablation studies are sufficiently comprehensive. Together, they provide strong evidence for the effectiveness of the proposed method.

### Weakness

1. The proposed method introduces several hyperparameters that require careful tuning ($B_0$, K, B, $\lambda$, $\beta$), which may hinder its practicality and adoption in real-world applications.
2. The experimental evaluation is somewhat limited: the authors consider only two Qwen base models of similar size, and all results are reported on a single math-focused dataset. Broader validation across more model families/sizes and more diverse tasks would strengthen the empirical claims.

---

> ### Author Rebuttal · Authors · 2026-03-30
>
> We sincerely thank the Reviewer `LWZc` for the positive assessment of our work and appreciate the constructive suggestions on practicality and broader validation. We provide our point-by-point responses below.
>
> > **Q1: Although the paper provides detailed ablation studies for each hyperparameter, it remains unclear whether the recommended settings would transfer to other base models or tasks (coding/agent). I would appreciate discussion or evidence on how sensitive these hyperparameters are to changes in model family, model scale, or task domain.**
>
> We fully agree that broader empirical validation would strengthen the paper. Following this suggestion, we conduct additional experiments using **recommended setting with original parameters**:
>
> 1. We additionally evaluate **Qwen3-14B-Base** to test whether the method scales beyond the two original Qwen base models:
>
> | Models | Math500 | AIME25 | AMC24 | GPQA | IFEval |
> |-|-|-|-|-|-|
> | Initial Model | 71.1% | 9.2% | 30.7% | 38.6% | 52.9% |
> | - Vanilla GRPO | 84.0% | 12.4% | 40.4% | 43.7% | 53.9% |
> | - GRPO w/ Entropy Reg | 81.9% | 17.2% | 45.4% | 44.2% | 52.5% |
> | - IBRO | 84.6% | 15.7% | 45.4% | 44.7% | 54.3% |
> | - TreeRL | 83.1% | 12.5% | 43.6% | 44.2% | 53.4% |
> | - IBTPO (Ours) | **84.8%** | **17.7%** | **47.4%** | **46.6%** | **57.9%** |
>
> 2. We additionally evaluate **Llama3.1-8B-Instruct** on a **code generation task** (trained on Open-R1 verifiable-coding-problems) to move beyond a single model family and beyond math-only task:
>
> |Models | LiveCodeBench | HumanEval | MBPP |
> |-|-|-|-|
> | Initial Model | 16.1% | 66.6% | 57.3% |
> | - Vanilla GRPO | 17.8% | 69.3% | 58.0% |
> | - IBTPO (Ours) | **19.5%** | **70.2%** | **64.8%** |
>
> These additional experiments further suggest that the effectiveness of IBTPO is not confined to the Qwen family or to math reasoning alone, and that the recommended setting shows promising transferability across model families, model scales, and task domains.
>
> We will include these new results in the revised paper.
>
> > **W1: The proposed method introduces several hyperparameters that require careful tuning (B_0, K, B, lambda, beta), which may hinder its practicality and adoption in real-world applications.**
>
> We agree that practicality is important, and we would like to clarify that these hyperparameters fall into two different categories:
>
> 1. Optimization hyperparameters $\lambda$ and $\beta$. We provide ablations for both, and our additional analysis suggests that the method is reasonably robust within practical ranges: (1) $\beta$ is insensitive around the recommended value.  In Table 3, $\beta=5$ performs best, while $\beta=1$ or $10$ remain competitive, indicating that the method does not require highly precise tuning. (2) Table 4 shows that the method is stable for moderate values of $\lambda$. While an overly large $\lambda$ can hurt performance, this is expected since it over-emphasizes IB-based term and may lead to over exploration. Importantly, the method is robust within a reasonable range (e.g., $0.05$ to $0.1$)
>
> 2. Tree-sampling hyperparameters $(B_0,K,B)$. They are not ad hoc tuning knobs introduced solely for performance, but are part of a general and unified formulation of the tree expansion process. Our motivation for using $(B_0,K,B)$ instead of a single scalar is to provide a flexible way to control different sampling regimes. This parameterization makes IBTree applicable to different computation/token budgets and different search preferences. Importantly, the empirical results in Table 3 and Table 5 suggest that **IBTree produces consistent better results across different sampling strategies**.
>
>
> > **W2: The experimental evaluation is somewhat limited: the authors consider only two Qwen base models of similar size, and all results are reported on a single math-focused dataset. Broader validation across more model families/sizes and more diverse tasks would strengthen the empirical claims.**
>
> We thank the reviewer for this suggestion. As discussed in Q1, we have now added experiments on Qwen3-14B-Base and Llama3.1-8B-Instruct for code generation task, which broaden the validation across model scale, model family and task domain. We will include these experiments in the revised paper.

---

> > ### Author Rebuttal · Reviewer_LWZc · 2026-04-01
> >
> > I thank the authors for the additional experimental results and explanations. My concerns have been largely resolved, and I will raise my score and confidence accordingly.

---

### Official Review · Reviewer_yuAp · 2026-03-11

**Soundness:** 2
**Presentation:** 3
**Significance:** 3
**Originality:** 3
**Overall Recommendation:** 5
**Confidence:** 2

**Summary:**

This paper proposes a novel metric grounded in Information Bottleneck theory that evaluates policy’s explorationexploitation balance. Based on it, the authors proposes  a principled framework that formulates IB-Score as a fine-grained optimization objective and utilizes a novel IB-guided tree sampling strategy

**Compliance With Llm Reviewing Policy:**

Affirmed.

**Key Questions For Authors:**

Could the authors provide more examples of the resulting search tree, as the one in Appendix D.3? This can help better understand the method.

**Strengths And Weaknesses:**

Pros:
1. The paper is well motivated and clearly written.
2. The proposed method is novel. IB-Score provides a theoretically grounded way to quantify reasoning diversity vs. correctness information, and IB-guided tree sampling strategy offers a structured alternative to standard trajectory sampling.
3. The method is theoretically grounded and experimental results also well support the effectiveness of the method.

Cons:
1. The framework (IB-Score + tree-based optimization) adds conceptual and implementation complexity, as well as algorithmic designs, compared with standard RL pipelines.
2. Some components of the framework (metric vs. sampling strategy) could be disentangled more clearly in ablation studies.

---

> ### Author Rebuttal · Authors · 2026-03-30
>
> We sincerely thank the Reviewer `yuAp` for the thorough assessment and valuable feedback. We provide our point-by-point responses below.
>
> > **Q1: Could the authors provide more examples of the resulting search tree, as the one in Appendix D.3? This can help better understand the method.**
>
> We appreciate this suggestion for improving the clarity of our method. We have prepared additional visualizations of IBTree results. Additional examples are available at anonymous repository: https://anonymous.4open.science/r/IBTree_Visualization-110B/
>
> We will incorporate these examples into Appendix D.3 in the revised paper.
>
> > **W1: The framework (IB-Score + tree-based optimization) adds conceptual and implementation complexity, as well as algorithmic designs, compared with standard RL pipelines.**
>
> We acknowledge that IB-TPO introduces additional components and complexity compared to vanilla GRPO, but this complexity is necessary and well-justified for the following reasons:
>
> 1. Tree-based sampling is a well-established and valuable research direction: Tree-structured reasoning has been extensively studied in LLM reasoning, demonstrating significant benefits for exploration efficiency and reasoning quality. The additional complexity brings substantial performance gains (2.9-3.6% improvement over GRPO baseline) and signigicant higher token efficiency (enabling 50% more trajectories under same token budget).
>
> 2. Modern inference engines (e.g. vLLM) provide optimized support for tree-based reasoning (e.g. prefix caching). Building upon this, we have implemented a highly optimized tree-sampling codebase to maximize parallelism and fully leverage the decoding performance of the inference engine. In future work, we will continue refining tree sampling to ensure its efficiency and simplicity.
>
> 3. We conducted thorough ablation studies on hyperparameters and experiments demonstrate that the default hyperparameters consistently yield stable performance gains across different model families, model sizes, and task domains, confirming their robustness (please refer to experiments in Table 1 and additional experiments to the Q1 of Reviewer `LWZc`). We also provide detailed reports in our experiments on the performance of IBTree under various parameter settings to assist practitioners in making hyperparameter choices.
>
> > **W2: Some components of the framework (metric vs. sampling strategy) could be disentangled more clearly in ablation studies.**
>
> We thank for this valuable suggestion. Our existing ablation studies (Tables 2,3,5) provide component-wise analysis:
>
> 1. Ablation study in Table 2 isolates the two main components: metric (IB-Score) and sampling strategy (IBTree) at the training level. The results show that both components independently improve performance, and their combination yields the strongest gains. This is intended to disentangle the contribution of the sampling strategy from that of the IB-based optimization signal.
>
> 2. Ablation study in Table 3 and Table 5 further isolate the role of the sampling strategy itself. These tables compare multiple branching strategies under the same tree-based framework, including random branching, fixed-width branching, entropy-guided branching, and IB-Score-guided branching. This is meant to distinguish the benefit of the tree structure alone from the benefit of using IB-Score as the branching guidance. The consistent advantage of IB-Score-guided branching suggests that the gain is not merely from having a tree, but from the IB-based guidance.
>
> To further disentangle the contribution of the IB-based optimization objective (IBTPO Adv) from that of our IB-guided sampling strategy (IBTree), we conducted additional experiments on RandTree (random branching) and EPTree (entropy-guided branching):
> |Models|AIME25|AMC24|GPQA|
> |-|-|-|-|
> |TreeRL (w/ EPTree)|14.9%|40.5%|39.8%|
> |Vanilla GRPO|13.6%|39.4%|38.1%|
> |+ RandTree & IBTPO Adv|14.5%|39.8%|37.3%|
> |+ EPTree & IBTPO Adv|15.0%|42.3%|40.9%|
> |+ IBTree & IBTPO Adv (Ours)|**15.3%**|**46.0%**|**41.7%**|
>
> where we can find: (1) EPTree & IBTPO Adv outperforms TreeRL, demonstrating that IB-based objective provides non-marginal improvement. (2) The full combination (IBTree & IBTPO Adv) achieves the best results, confirm that IBTree with IB-Score guidance cannot be replaced by alternative tree structures.
> We will include these additional ablations in the revised paper to make the disentanglement between metric and sampling strategy more explicit.

---

> > ### Author Rebuttal · Reviewer_yuAp · 2026-04-01
> >
> > Thanks for the authors response. My concerns are addressed.

---

### Official Review · Reviewer_J2YP · 2026-03-12

**Soundness:** 3
**Presentation:** 3
**Significance:** 3
**Originality:** 4
**Overall Recommendation:** 5
**Confidence:** 4

**Summary:**

This paper studies the exploration-exploitation imbalance in online reinforcement learning for large language models. It argues that existing methods such as GRPO can struggle to maintain an effective balance during training, which may lead either to premature convergence and weak exploration or to excessive uncertainty and unstable optimization. To analyze this issue, the paper introduces IB-Score, a fine-grained metric motivated by Information Bottleneck theory, to measure the balance between reasoning diversity and answer-relevant information during multi-step reasoning. Building on this perspective, the paper proposes IB-TPO, a tree-based policy optimization framework that incorporates IB-Score into training. Its core idea is to use an IB-guided tree search (IBTree) to selectively expand promising reasoning branches, both to improve sampling efficiency and to support Monte Carlo estimation of the IB-based objective. The resulting framework combines structured exploration with step-level advantage estimation for policy optimization. The paper evaluates the method on Qwen3 models across several reasoning benchmarks and reports consistent improvements over GRPO and other online RL baselines.

**Compliance With Llm Reviewing Policy:**

Affirmed.

**Final Justification:**

I think this article is worth reading; it has a clear purpose and a logical structure.

**Key Questions For Authors:**

**1. Could the authors clarify how they would like readers to interpret the relative roles of the IB-based objective and the tree-based sampling structure in Table 2?**

The ablation already suggests that both components contribute, but the relative importance seems to vary across benchmarks, and in some cases IBTree appears to account for a large share of the gain. A clearer discussion of what the IBTPO advantage adds beyond IBTree, and why the full combination is necessary, would strengthen the paper’s central claim.

**2. Should IB-Score be viewed as a principled derivation from the Information Bottleneck objective, or as a practical surrogate motivated by it?**

I found the intuition behind IB-Score compelling, but the method also relies on several approximations and estimation choices. A clearer explanation of which parts are theoretically essential and which are mainly practical would increase my confidence in the soundness of the paper and help me judge the contribution more accurately.

**Limitations:**

yes

**Strengths And Weaknesses:**

On soundness, I find the paper promising but not fully airtight. The main technical concern is that the central theoretical object, IB-Score, is motivated persuasively but not yet justified as rigorously as I would like. The derivation moves from the IB objective to a step-level surrogate through several approximations and estimation choices, including Monte Carlo estimates of reward density and a reformulation whose practical interpretation is plausible but not obviously unique or inevitable. This does not make the method unsound, but it does mean that the paper reads more as a well-motivated objective construction than as a fully established principled derivation. In addition, while the experiments show consistent gains, it remains somewhat unclear how much of the improvement comes specifically from the IB-based objective and how much comes from the tree-based sampling structure itself. The ablation suggests that IBTree alone already contributes substantially, while the full method performs best but by a more modest margin.

A second soundness-related issue is fairness and cost comparability. The paper makes a real effort here by aligning experimental settings and matching tree-based and independent sampling approximately by token budget, which is a genuine strength. However, the comparison is still not entirely clean. Some baselines are re-tuned, some are reimplemented by the authors due to lack of official code, and the tree-based methods introduce a different sampling structure that may itself be a large part of the gain. Moreover, the paper explicitly acknowledges that IBTree remains somewhat slower than independent sampling in wall-clock time despite token-efficiency benefits, which is important because the method’s practical value depends on the full cost-performance tradeoff rather than token count alone.

On presentation, the paper is generally clear and well organized. The high-level story is easy to follow, the method overview figure is helpful, and the progression from diagnosis to algorithm to experiments is logical. I did not find major readability problems. My concerns here are relatively minor: some claims are phrased somewhat strongly, especially when describing the superiority or unifying role of the proposed framework, and the theoretical section is fairly dense because several approximations and quantities are introduced in a short span.

In terms of significance, I view the paper positively. The paper addresses it in a way that is relevant to ongoing work on RLVR and reasoning-oriented post-training. Even if the contribution is somewhat specialized to online RL for reasoning models, it still has meaningful potential impact: the IB-Score perspective could influence how researchers diagnose training dynamics, and the tree-based optimization framework offers a practical recipe for improving sampling effectiveness under constrained budgets. The empirical gains are not enormous, but they are consistent, and the work is likely to be of interest to researchers building training pipelines for reasoning models.

On originality, I think the paper has moderate but real novelty. The individual ingredients are not all new: information-bottleneck ideas have already been explored in RL-style optimization, and tree-based sampling for LLM reasoning is also an active area. The originality here comes from the combination: the paper proposes a step-level IB-inspired metric, uses it both as a diagnostic lens and as a guide for branching and advantage estimation, and integrates these pieces into a unified online RL framework. I do not see this as a fundamentally new theoretical paradigm, but I do think it is a meaningful and reasonably creative synthesis that yields useful empirical insight. In that sense, the paper’s originality lies more in the perspective and system design than in a sharply novel standalone algorithmic primitive.

So overall, I see this as a strong and interesting paper with credible empirical value, but with remaining questions around the exact role of IB-Score, the degree of theoretical grounding, and the extent to which the gains should be attributed to the new objective rather than to the tree-based sampling structure.

---

> ### Author Rebuttal · Authors · 2026-03-30
>
> We sincerely thank the Reviewer `J2YP` for the thorough and constructive feedback. We provide our point-by-point responses below.
>
> > **Q1: Could the authors clarify how they would like readers to interpret the relative roles of the IB-based objective and the tree-based sampling structure in Table 2?**
>
> We appreciate this insightful question. The relationship between IB-based objective and the tree-based sampling structure is **synergistic rather than independent**:
>
> 1. IB-based objective (IB-Score) provides a fine-grained learning signal grounded in IB theory.
>
> 2. Tree-based sampling structure (IBTree) serves as an efficient structure for RL rollouts and IB-Score estimation.
>
> Table 2 shows that IBTree or IBTPO adv alone already improves over GRPO baseline, while the full IBTPO performs best. This pattern reflects the **mutual reinforcement** between the two components: IBTree benefits from IB-Score guidance for selective branching (Table 3 shows IB-Score guided branching outperforms other strategies), while the IB-based objective relies on IBTree for efficient estimation and structured exploration. Thus, the observation that IBTree contributes substantially does not diminish the importance of IB-Score, rather, it demonstrates that IBTree effectively leverages IB-Score guidance.
>
> To provide a more complete ablation and more comprehensively disentangle the roles of the tree structure and IB-Score guidance, we conducted additional experiments on RandTree (random branching) and EPTree (entropy-guided branching):
> |Models|AIME25|AMC24|GPQA|
> |-|-|-|-|
> |TreeRL (w/ EPTree)|14.9%|40.5%|39.8%|
> |Vanilla GRPO|13.6%|39.4%|38.1%|
> |+ RandTree & IBTPO Adv|14.5%|39.8%|37.3%|
> |+ EPTree & IBTPO Adv|15.0%|42.3%|40.9%|
> |+ IBTree & IBTPO Adv (Ours)|**15.3%**|**46.0%**|**41.7%**|
>
>
> where we can find: (1) EPTree & IBTPO Adv outperforms TreeRL, demonstrating that IB-based objective provides non-marginal improvement. (2) The full combination (IBTree & IBTPO Adv) achieves the best results, confirm that IBTree with IB-Score guidance cannot be replaced by alternative tree structures, reinforcing the synergistic nature of our framework.
>
> > **Q2: Should IB-Score be viewed as a principled derivation from the Information Bottleneck objective, or as a practical surrogate motivated by it?**
>
> Thank you for this important question. Our intended interpretation is that **IB-Score should be viewed primarily as a principled derivation from the Information Bottleneck objective, with necessary practical adaptations for online RL setting**. Its core formulation directly follows IB theory's mathematical definition (Eq.3) and the step-wise form (Eq.5) is derived through standard information-theoretic expansions, which are both theoretically essential components. We also introduces necessary and principled adaptations for applying IB theory to LLM online RL: (1) We use Tsallis entropy (Eq.6 and 7) as a deliberate choice to address the numerical instability by logarithmic calculation of Shannon entropy. (2) We apply Monte Carlo methods to estimate step-level entropy and reward density, which is an essential and widely used **unbiased** technique for handling intractable full distributions.
>
> In addition to the two direct questions above, we further address two broader concerns raised implicitly in the review below.
>
> > **W1: Some baselines are re-tuned, some are reimplemented by the authors due to lack of official code.**
>
> We try our best to ensure the fairness and reliability of our experiments. For parameters requiring re-tuning, we perform meticulous adjustments and select the optimal settings for comparison. Regarding IBRO, although official code is unavailable, we strictly implemented it based on the algorithms provided in the official paper. Furthermore, we provide detailed specifications for all compared methods in the Appendix.B.2 to ensure reproducibility.
>
> > **W2: Moreover, the paper explicitly acknowledges that IBTree remains somewhat slower than independent sampling in wall-clock time despite token-efficiency benefits, which is important because the method’s practical value depends on the full cost-performance tradeoff rather than token count alone.**
>
> We agree that under the experimental settings, IBTree exhibits a slightly longer wall-clock runtime compared to independent sampling. We consider this a common limitation of current tree-based methods and a key direction for future work. It is important to note that: (1) Benefiting from optimizations within the inference engine (e.g., prefix caching), the runtime overhead introduced by multi-iteration expansion is effectively controlled, preserving its practical value; (2) Our current tree sampling implementation still has room for improvement. In our pilot implementation, asynchronous expansion across different trees can further reduce inter-iteration waiting, yielding about 15% runtime reduction. Please refer reponses to W1&W2 of the Reviewer `uzxw` for detailed experiments.

---

> > ### Author Rebuttal · Reviewer_J2YP · 2026-04-04
> >
> > The content and quality of this article are both very high, making it highly convincing. The author's responses to these questions were also very good.

---

### Decision · Program_Chairs · 2026-04-30

**Decision:**

Accept (regular)

**Comment:**

The authors propose a principled framework that formulates IB-Score (information bottleneck) as a fine-grained optimization objective and utilizes a novel IB-guided tree sampling strategy. The authors addressed most of the concerns during the rebuttal, and all reviewers are positive about the paper.